

# Assessing Lagrangian Coherence in Atmospheric Blocking

Henry Schoeller[1], Robin Chemnitz[2], Péter Koltai[3], Maximilian Engel[4,2], and Stephan Pfahl[1]

[1]Institute of Meteorology, Freie Universität Berlin, Berlin, Germany
[2]Mathematics Institute, Freie Universität Berlin, Berlin, Germany
[3]Faculty of Mathematics, Physics & Computer Science, Universität Bayreuth, Bayreuth, Germany
[4]Institute of Mathematics Korteweg-De Vries, University of Amsterdam, Amsterdam, Netherlands

**Correspondence:** Henry Schoeller (henry.schoeller@fu-berlin.de)

**Abstract.** Atmospheric blocking exerts a major influence on mid-latitude atmospheric circulation and is known to be associated with extreme weather events. Previous work has highlighted the importance of the origin of air parcels that define the blocking region, especially with respect to non-adiabatic processes such as latent heating. So far, an objective method of clustering the individual Lagrangian trajectories passing through a blocking into larger and, more importantly, spatially coherent air streams
has not been established. This is the focus of our study.

To this end, we determine coherent sets of trajectories, which are regions in the phase space of dynamical systems that keep their geometric integrity in time, and can be characterized by robustness under small random perturbations. We approximate a dynamic diffusion operator on the available Lagrangian data and use it to cluster the trajectories into coherent sets. Our implementation adapts the existing methodology to the non-Euclidean geometry of Earth's atmosphere and its challenging
scaling properties. The framework also allows statements about the spatial behavior of the trajectories as a whole. We discuss two case studies differing with respect to season and geographic location.

The results confirm the existence of spatially coherent feeder air streams differing with respect to their dynamical properties and, more specifically, their latent heating contribution. Air streams experiencing a considerable amount of latent heating occur mainly during the maturing phase of the blocking and contribute to its stability. In our example cases, trajectories also
exhibit an altered evolution of general coherence when passing through the blocking region, which is in line with the common understanding of blocking as a region of low dispersion.

## 1 Introduction

Atmospheric blocking represents a critical phenomenon in the dynamics of Earth's atmosphere, characterized by the temporary obstruction of prevailing westerlies in the mid-latitudes, potentially influencing weather on a planetary scale (Lupo, 2021).
These events are notable for their role in causing extreme weather conditions, such as heatwaves, cold spells, and sustained periods of precipitation, impacting both human activities and natural ecosystems (Kautz et al., 2022; Pfahl and Wernli, 2012). The mechanisms leading to atmospheric blocking are complex, involving interactions between the atmosphere, cryosphere (Tyrlis et al., 2019; Woollings et al., 2014), ocean (Drijfhout et al., 2013; Häkkinen et al., 2011), and land (He et al., 2014; Kurgansky, 2020; Tilly et al., 2008), and are a subject of ongoing research. Despite their significance, predicting atmospheric





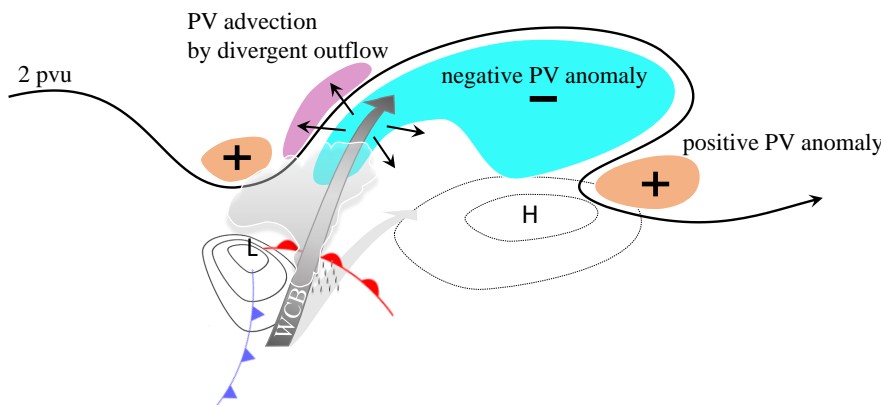

**Figure 1.** Schematic depicting a WCB air stream stabilizing a high pressure blocking, which is depicted as an upper-level negative PV anomaly. Reproduced from Steinfeld (2019) with permission.

blocking events and their impacts remains a challenge, due to the inherent variability and the multifaceted processes that govern their formation, maintenance, and dissipation. Weather and climate models' inability to correctly represent blocking (Matsueda, 2009) also causes considerable uncertainties in its predicted response to climate change (Woollings et al., 2018).

Though traditionally conceived as an adiabatic phenomenon, recent studies have pointed to the role of diabatic processes in blockings (Croci-Maspoli and Davies, 2009; Hauser et al., 2023; Pfahl et al., 2015; Tilly et al., 2008). More specifically, a
line of study has been concerned with moist processes, arguing that a considerable fraction of the air masses constituting the blocking originate from warm sectors of neighboring surface cyclones and travel into the blocking via warm conveyor belts (WCBs) (Pfahl et al., 2015; Steinfeld and Pfahl, 2019; Yamamoto et al., 2021). WCBs are moist, rapidly ascending air streams and are subject to research in their own right (e.g., Joos et al., 2023; Madonna et al., 2014) i.a. due to their contribution to forecast uncertainty (Madonna et al., 2015; Pickl et al., 2023; Wandel et al., 2021). According to Steinfeld and Pfahl (2019), the
air parcels conveyed by such WCBs considerably influence the blocking by stabilizing and potentially intensifying the negative potential vorticity (PV) anomaly characteristic for the blocking especially in the onset and maintenance stages of its lifecycle. This happens, firstly, through a material change in PV of the respective air masses via latent heating induced, cross-isentropic vertical motion and, secondly, through the divergent outflow of the guiding air streams in the upper troposphere. Evidence that moist processes might become more important in a warmer and moister climate underlines the need for further investigation
(Steinfeld et al., 2022). A schematic depicting the procedural connections is reproduced from Steinfeld (2019) in Fig. 1.

The notion of a WCB suggests a geometrically coherent flow of the air masses involved, but such a coherent behavior has not been explicitly addressed in the studies above. The backward trajectories attributed to such air streams are typically identified using thresholds for ascent (change in pressure) or diabatic heating (change in potential temperature) such that a common pathway is not guaranteed. In addition, the methodology employed does not allow for statements about the size, location,
time of existence or even the number of WCBs implicated in the process. Another drawback of the identification method is its subjective nature implied by selecting an arbitrary threshold for the decrease in pressure (Madonna et al. (2014) use 600





hPa in 48 h) or increase in potential temperature the air parcels have experienced (usually a threshold of $\Delta\theta > 2$ K is used). To the best of our knowledge, an objective identification method for coherent medium-to-large scale air streams that allows addressing these drawbacks has not been presented in the context of atmospheric sciences.

In this study, we want to tackle these issues focusing on the exact spatio-temporal nature of trajectories passing through a blocking and how individual trajectories align with each other. By grouping the set of trajectories into subsets (clusters) based on only their geometric coherence, we identify synoptic scale air streams and analyze their individual dynamical properties and interrelation with the large scale flow configuration. The perspective introduced also allows for statements about the overall spatial configuration and coherence of the air parcels traced, especially with respect to the blockings' lifecycles. A blocking's

stabilizing and dispersion-suppressing nature affects individual air parcels, which can be identified through our Lagrangian lens.

   To accomplish the above, we make use of the mathematical concept of coherent sets. A coherent set is a time-evolving region in the state space of a dynamical system that keeps its geometric integrity to a large degree. This is particularly interesting when the underlying system is such that individual trajectories diverge and any set eventually becomes highly filamented under

evolution through the dynamics. We characterize coherent sets based on their stability under small perturbations. Coherent sets hence resist dispersion, persist longer in complicated flows and have thus an outstanding effect on the transport of physically relevant quantities. The fundamental idea is that in a coherent set the individual trajectories stay relatively close together as a whole, such that particles remain within the coherent set when they are subjected to small random perturbations. In contrast, if a time-evolving set is filamented, particles are likely to leave the set under the influence of random noise – and hence mix

with its exterior. To formalize this heuristics and numerically compute coherent sets, we follow the work of Banisch and Koltai (2017) (based on theory developed in Froyland (2013, 2015)), which characterizes the robustness of coherent sets under small perturbations using a data-analysis framework called diffusion maps. Their method yields a single time-averaged operator whose spectral properties contain the necessary information to extract coherent sets.

   The visual idea of coherence (i.e., "trajectories staying together") can be cast into diverse objectives, which then can be

used to partition available trajectory data into such sets. For a sample of the ideas that have been implemented please consult Froyland and Padberg-Gehle (2015); Allshouse and Peacock (2015); Hadjighasem et al. (2016); Schlueter-Kuck and Dabiri (2017); Padberg-Gehle and Schneide (2017); Koltai and Renger (2018). A related notion is that of Lagrangian coherent structures (Haller and Beron-Vera, 2013; Haller, 2015), aiming to describe barriers of transport. For their computation from finite trajectory data, see for instance Mowlavi et al. (2022).

The main foci of this study are the adaptation of the methodological framework of Banisch and Koltai (2017) to real world trajectory data of air parcels in the atmosphere and its deployment for case studies of atmospheric blockings. This is motivated by the question of spatial coherence in WCBs in the context of blocking. As such, the presented work does not endeavour to give a one-size-fits-all solution to the problem of finding coherent air streams regardless of scale, geographic location and synoptic condition, but should rather be understood as a proof of concept. We think atmospheric blocking is a phenomenon well

suited for the application of the the developed methodology, since it is large enough to feature a handful of distinct, coherent air streams (given the resolution of our data), but small enough to be well resolved by the number of trajectories computationally





feasible. We have thoroughly analyzed two cases of atmospheric blocking differing with respect to both geographical location and time of year and show that differences between the two examples at similar points in their lifecycles are notably smaller than differences within the same example between different points in the lifecycles. This goes for the occurrence of WCBs, the

overall trajectory density and the traced air masses' filamentation.

The paper is organized as follows. Sect. 2 introduces the mathematical foundation relevant to the methods employed. Sect. 3 covers details about the implementation of our algorithm as well as the data used. We show results from the application of the algorithm to test cases in Sect. 4 to both elucidate the concepts and methods introduced before and investigate the problems and questions posed above. Finally, Sect. 5 summarizes the findings.

## 90 2 Theory

To extract coherent sets from Lagrangian trajectory data, we employ the method proposed by Banisch and Koltai (2017). It uses local distances between data points to construct a diffusion operator that is used to estimate coherent sets by performing spectral clustering. In the following, we give a brief introduction into coherent sets and diffusion maps. Our aim is to provide a good intuitive understanding. For a formal and mathematically rigorous introduction of coherent sets using a transfer–operator

based framework, we refer to Banisch and Koltai (2017) and Froyland (2013). For a formal introduction to diffusion maps, we refer to Coifman and Lafon (2006).

### 2.1 Coherent sets

Informally put, coherent sets are time-dependent regions in space that remain largely intact as the system evolves. They exhibit a "natural" separation from their surrounding in the sense that flow across the boundary is small and geometric integrity is kept

to a large degree. We characterize the geometric integrity of coherent sets by their robustness with respect to the addition of small noise. We first introduce the heuristics of coherent sets in an abstract setting before proceeding to the computation of coherent sets from trajectory data.

Consider the evolution of points over a finite number of time steps in a non-autonomous dynamical system. Let $\mathbb{X}_t \subset \mathbb{R}^n$ be bounded sets which denote the domain of the dynamical system at each point in time. Here, $t$ ranges over the integers from

0 to $T$. A non-autonomous dynamical system in discrete time over these time-evolving domains is given by a sequence of bijective maps $\Phi_{t+1,t} : \mathbb{X}_t \to \mathbb{X}_{t+1}$, where $0 \le t \le T-1$. Before characterizing coherent sets of the dynamical system over the entire time frame from 0 to $T$, we study sets that are coherent under a single step.

Consider a bijective map $\Phi : \mathbb{X} \to \mathbb{Y}$, with bounded domains $\mathbb{X}, \mathbb{Y} \subset \mathbb{R}^n$. We say that a set $\mathbb{A} \subset \mathbb{X}$ is coherent under $\Phi$ if the relation $(\Phi^{-1} \circ \Phi)(\mathbb{A}) = \mathbb{A}$ is robust under the addition of small noise both at initial and final time. As a consequence, we require

that the sets $\mathbb{A}$ and $\Phi(\mathbb{A})$ are not too filamented, i.e. that they possess high geometric integrity. Let $\mathcal{D}_\epsilon$ be a diffusion operator that applies an $\epsilon$-small random perturbation to any given point. The domain which $\mathcal{D}_\epsilon$ acts on will be clear from the context, i.e. $\mathcal{D}_\epsilon : \mathbb{X} \to \mathbb{X}$ or $\mathcal{D}_\epsilon : \mathbb{Y} \to \mathbb{Y}$. A coherent set $\mathbb{A} \subset \mathbb{X}$ of the map $\Phi$ has to satisfy $\mathcal{D}_\epsilon \mathbb{A} \approx \mathbb{A}$ as well as $(\Phi^{-1} \circ \mathcal{D}_\epsilon \circ \Phi)(\mathbb{A}) \approx \mathbb{A}$. This heuristics is formulated more precisely in the language of transfer operators in Banisch and Koltai (2017).



The above heuristics describe coherent sets of a single map $\Phi$. We return to the setting of a non-autonomous system over

$T$ time steps given by the maps $\Phi_{t+1,t}$. A coherent set is a family of sets $(\mathbb{A}_t)_{0 \leq t \leq T}$, where $\mathbb{A}_t \subset \mathbb{X}_t$ and $\Phi_{t+1,t}(\mathbb{A}_t) = \mathbb{A}_{t+1}$. Define the maps $\Phi_{t,0} = \Phi_{t,t-1} \circ \ldots \circ \Phi_{1,0}$. For $t = 0$, the map $\Phi_{0,0}$ is simply the identity. A coherent set is completely characterized by $\mathbb{A}_0$, in the sense that $\mathbb{A}_t = \Phi_{t,0}(\mathbb{A}_0)$. We say that the family $(\mathbb{A}_t)_{0 \leq t \leq T}$ is coherent if

$$(\Phi_{t,0}^{-1} \circ \mathcal{D}_\epsilon \circ \Phi_{t,0})(\mathbb{A}_0) \approx \mathbb{A}_0, \tag{1}$$

for all $t$ from 0 to $T$. We remark that this concept can be generalized to continuous time systems, e.g. Matthes et al. (2016).

Note that we assumed that $\mathcal{D}_\epsilon$ maps from $\mathbb{X}_t$ into $\mathbb{X}_t$, i.e. the random perturbation cannot map points outside of the domain $\mathbb{X}_t$. This corresponds to e.g. reflecting boundary conditions. For the purposes of our application, reflecting boundary conditions are physically not justifiable, since there are no physical boundaries for particles to be reflected at. Instead, we assume that $\mathcal{D}_\epsilon$ can transport points outside of $\mathbb{X}_t$. Since we assume no information about the dynamics outside of $\mathbb{X}_t$, we consider points that are mapped outside of $\mathbb{X}_t$ as lost and hence unable of contributing to coherence. This corresponds to absorbing

boundary conditions. We discuss how absorbing boundary conditions are implemented in the next subsection.

## 2.2  Diffusion maps

In the following, we give a brief introduction to diffusion maps. A mathematically rigorous derivation can be found in Coifman and Lafon (2006) and Ghojogh et al. (2022).

Assume the data of $m$ trajectories of a non-autonomous dynamics is provided in the form of data-points $x_t^i \in \mathbb{R}^n$, where

$1 \leq i \leq m$ and $t$ ranges over the integers from 0 to $T$. We assume that in each time frame the point cloud $X_t = \{x_t^i \,|\, 1 \leq i \leq m\}$ approximates a bounded manifold $\mathbb{X}_t \subset \mathbb{R}^n$ and that the data-points are sample trajectories of a – potentially unknown – non-autonomous dynamics $\Phi_{t+1,t} : \mathbb{X}_t \to \mathbb{X}_{t+1}$, i.e. $\Phi_{t,s}(x_s^i) = x_t^i$ for all $s \leq t$. Our goal is to characterize coherent sets of this dynamics, in particular to formalize the heuristics Eq. (1) in the setting of discrete trajectory data. A coherent set consists of a certain subset of the point cloud, i.e. $A_t := \{x_t^i \,|\, i \in \mathcal{I}\} \subset X_t$, where $\mathcal{I} \subset \{1, \ldots, m\}$ be a set of indices. In order to formalize

the heuristic in Eq. (1), we introduce a discretization of the operator $\mathcal{D}_\epsilon : \mathbb{X}_t \to \mathbb{X}_t$ acting on the point cloud $X_t$.

Since $X_t$ is a finite set of size $m$, we construct a transition probability matrix $\hat{\mathbf{P}}_{t,\epsilon} \in \mathbb{R}^{m \times m}$ that simulates diffusion on $X_t$ of strength $\epsilon$. The rest of this section is devoted to constructing the matrix $\hat{\mathbf{P}}_{t,\epsilon}$ for each $0 \leq t \leq T$, as well as discussing the implementation of boundary conditions. We remark that we do not need to approximate the dynamics $\Phi_{t+1,t}$ to use Eq. (1), since we assume to have the data already in the form of trajectories.

Let $k_\epsilon$ be a symmetric diffusion kernel with scale parameter $\epsilon$. This kernel encodes the proximity of two data-points, i.e. $k_\epsilon(x^i, x^j)$ is large if $x^i$ and $x^j$ are close and small if the points are far apart. In the following, we use the Gaussian kernel for an arbitrary metric $\text{dist}(\cdot, \cdot)$ on $\mathbb{R}^n$,

$$k_\epsilon(x^i, x^j) = \exp\left(-\frac{\text{dist}(x^i, x^j)^2}{\epsilon}\right). \tag{2}$$

In general, it is natural to use the Euclidean distance $\text{dist}(x^i, x^j) = \|x^i - x^j\|$. However, for the application to atmospheric

data, which is a non-isotropic setting, we are going to use an alternative distance aiming to establish isotropy with respect to



turbulent diffusion, cf. Sect. 3.4. For each time step $0 \le t \le T$, we construct the similarity matrix $\mathbf{K}_{t,\epsilon} \in \mathbb{R}^{m \times m}$,

$$\mathbf{K}_{t,\epsilon}(i,j) = k_\epsilon(x_t^i, x_t^j). \tag{3}$$

Note that $\mathbf{K}_{t,\epsilon}$ is symmetric, all diagonal entries are 1, while all off-diagonal entries are between 0 and 1. In practice, a cutoff radius is used to increase the sparsity of the matrix by setting entries $\mathbf{K}_{t,\epsilon}(i,j)$ that are below a specified threshold to 0. Since

the value $\mathbf{K}_{t,\epsilon}(i,j)$ decays monotonically in the distance between $x_t^i$ and $x_t^j$, we may equivalently choose a cutoff radius $r$ and set $\mathbf{K}_{t,\epsilon}(i,j)$ to 0 if $\mathrm{dist}(x_t^i, x_t^j) > r$. Here, we choose $r = 3\sqrt{\epsilon}$ which is an appropriate scaling for a Gaussian kernel $k_\epsilon$.

To account for differences in the density of the point cloud, we pre-normalize the similarity matrix:

$$u_{t,\epsilon}(i) := \sum_{j=1}^m \mathbf{K}_{t,\epsilon}(i,j), \quad \hat{\mathbf{K}}_{t,\epsilon}(i,j) := \frac{\mathbf{K}_{t,\epsilon}(i,j)}{u_{t,\epsilon}(i)u_{t,\epsilon}(j)}. \tag{4}$$

Finally, the diffusion matrix $\hat{\mathbf{P}}_\epsilon$ is obtained by row-normalization

$$v_{t,\epsilon}(i) := \sum_{j=1}^m \hat{\mathbf{K}}_{t,\epsilon}(i,j), \quad \hat{\mathbf{P}}_{t,\epsilon}(i,j) := \frac{\hat{\mathbf{K}}_{t,\epsilon}(i,j)}{v_{t,\epsilon}(i)}. \tag{5}$$

The matrix $\hat{\mathbf{P}}_{t,\epsilon}$ is non-negative and normalized such that it is row-stochastic (often called left stochastic), i.e. the entries of each row sum to 1. Hence, $\hat{\mathbf{P}}_{t,\epsilon}$ can be understood as transition probabilities on the point cloud $\{x_t^i \mid 1 \le i \le m\}$ that simulate a discretized diffusion on the point cloud $X_t$. In the data-rich limit, $\hat{\mathbf{P}}_{t,\epsilon}$ is a self-adjoint operator, i.e. a symmetric matrix.

Lastly, we address the implementation of boundary conditions. Since $\hat{\mathbf{P}}_{t,\epsilon}$, as constructed above, is a stochastic matrix, all

points in $X_t$ remain in $X_t$, i.e. there are reflecting boundary conditions. However, in the context of atmospheric flow of air masses, where $\mathbb{X}_t$ is a bounded, time-dependent region of the atmosphere, it is unnatural to assume turbulent diffusion would not act across the boundaries of $\mathbb{X}_t$. Since we assume no information about the dynamics outside of $\mathbb{X}_t$, we assume absorbing boundary conditions, i.e. all points on the boundary of $X_t$ are removed. Hence, we need to determine a set of boundary points $\partial X_t \subset X_t$. Determining these point algorithmically is the content of Sect. 2.5. Given a set of boundary points $\partial X_t$, we

integrate absorbing boundary conditions into the transition matrix $\hat{\mathbf{P}}_{t,\epsilon}$ by removing the rows and columns corresponding to the boundary points. In order to keep the dimensions of the matrices compatible across different time steps, we implement this by setting the respective rows and columns to 0 instead of removing them:

$$\mathbf{P}_{t,\epsilon}(i,j) := \begin{cases} 0, & x_t^i \in \partial X_t, \text{ or } x_t^j \in \partial X_t, \\ \hat{\mathbf{P}}_{t,\epsilon}(i,j), & \text{else}. \end{cases} \tag{6}$$

By construction, $\mathbf{P}_{t,\epsilon}$ is a substochastic matrix. In summary, $\hat{\mathbf{P}}_{t,\epsilon}$ corresponds to a discretized diffusion matrix with reflecting

boundary conditions while $\mathbf{P}_{t,\epsilon}$ corresponds to a discretized diffusion matrix with absorbing boundary conditions. For the purposes of our application, we will go forward using the substochastic matrix $\mathbf{P}_{t,\epsilon}$.

## 2.3 Spectral clustering

Having constructed the diffusion matrices $\mathbf{P}_{t,\epsilon}$, we describe how to compute coherent sets using a spectral clustering method. A coherent set $A_t$, where $0 \le t \le T$, is given by $A_t = \{x_t^i \mid i \in \mathcal{I}\}$, where $\mathcal{I} \subset \{1, \ldots, m\}$ be a set of indices. In particular, we





find $\Phi_{t,0}A_0 = A_t$. Hence, the heuristic in Eq. (1), using the diffusion matrix $\mathbf{P}_{t,\epsilon}$ introduced in the previous section, requires
that indices $i \in \mathcal{I}$ have a high transition probability to $\mathcal{I}$, i.e.

$$\mathbf{P}_{t,\epsilon}(i, \mathcal{I}) := \sum_{j \in \mathcal{I}} \mathbf{P}_{t,\epsilon}(i,j) \approx 1,$$

for all $i \in \mathcal{I}$. This approximation should be as accurate as possible for all $0 \le t \le T$. We construct the averaged diffusion
operator

$$\mathbf{Q}_\epsilon := \frac{1}{T+1} \sum_{t=0}^{T} \mathbf{P}_{t,\epsilon}. \tag{7}$$

This averaged diffusion operator was introduced in Froyland (2015) and it was shown that coherent sets can be extracted from
the dominant eigenvectors of $\mathbf{Q}_\epsilon$. Equation (7) should be understood as a quantitative version of averaging the left-hand side
of Eq. (1). Thus, eigenvectors of $\mathbf{Q}_\epsilon$ for eigenvalues close to 1 represent functional representations of sets that satisfy Eq. (1)
on average for $0 \le t \le T$.

To better understand how coherent sets are related to the eigenvectors of $\mathbf{Q}_\epsilon$, assume that there are $K$ idealized coherent sets
$A_t^k$, for $1 \le k \le K$, corresponding to the sets of indices $\mathcal{I}^k$. These sets are idealized coherent sets in the sense that $\mathbf{Q}_\epsilon(i, \mathcal{I}^k) = 1$
for all $i \in \mathcal{I}^k$. Since $\mathbf{Q}_\epsilon$ is substochastic, this implies that $\mathbf{Q}_\epsilon$ has a block-diagonal structure with blocks indicated by the sets $\mathcal{I}^k$.
In particular, for each $1 \le k \le K$, the matrix $\mathbf{Q}_\epsilon$ has an eigenvector with eigenvalue 1 that is supported only on the set $\mathcal{I}^k$.
Hence, the $K$ coherent sets $\mathcal{I}^k$ can be extracted from eigenvectors to the $K$ largest eigenvalues. Additionally, there is an
$(K+1)$-st set, which is the complement of the union of the $\mathcal{I}^k$ which we call the boundary set. The temporal evolution of $\mathbb{X}_t$
and, equivalently $X_t$, implies that $\partial X_t$ is time-dependent as well such that this boundary set is not necessarily equal to $\partial X_t$.
The boundary set, together with the coherent sets $\mathcal{I}^k$ form a partition of the set of all points $\{1, \ldots, m\}$.

Returning to the general setting, we compute the eigenvalues of the matrix $\mathbf{Q}_\epsilon$. In the data-rich limit, the matrix $\mathbf{Q}_\epsilon$ is
symmetric and, thus, only has real eigenvalues. If complex eigenvalues occur numerically, we discard the imaginary part and
only consider their real part. We order the eigenvalues from large to small. Since $\mathbf{Q}_\epsilon$ is substochastic, all eigenvalues lie between
0 and 1. We say that there is a *spectral gap* after the $K$-th eigenvalue, if the $(K+1)$-st eigenvalue is significantly smaller that
the first $K$ eigenvalues. We call these first $K$ eigenvalues the *dominant spectrum* and the corresponding eigenvectors the
*dominant eigenvectors*. After identifying a spectral gap, we perform a $k$-means clustering of the set of points $\{1, \ldots, m\}$ using
the information of the dominant eigenvectors. Assuming that the dominant spectrum consists of $K$ eigenvalues, each point
in $\{1, \ldots, m\}$ has $K$ characteristic values, namely the entries of the $K$ dominant eigenvectors. Using the $k$-means algorithm,
we group the $m$ points into $K+1$ clusters $\mathcal{I}^k$, for $1 \le k \le K+1$. Like motivated by the idealized setting, $K$ of these sets
correspond to coherent sets, while there is an additional $(K+1)$-st boundary set. See Fig. 5 for an example of the spectrum of
$\mathbf{Q}_\epsilon$ as well as the spectral clustering of the point cloud. The boundary set is colored in gray.

## 2.4 Choosing $\epsilon$

Recall the definition of the similarity matrix $\mathbf{K}_{t,\epsilon} \in \mathbb{R}^{m \times m}$ in Eq. (3) with the diffusion kernel introduced in Eq. (2), where
$\text{dist}(\cdot, \cdot)$ is a metric on $\mathbb{R}^n$. Under the assumption that the points $x_t^i$ are distributed uniformly with respect to $\text{dist}(\cdot, \cdot)$, it is



argued in Appendix A.2 of Koltai and Weiss (2020) and following Berry and Harlim (2016); Coifman et al. (2008) that $\epsilon > 0$ should be chosen, if possible, such that the following approximation is valid:

$$S_t(\epsilon) := \sum_{i,j} \mathbf{K}_{t,\epsilon}(i,j) \approx \sum_{i=1}^{m} C \int_{\mathbb{R}^{d(t)}} \exp\left(-\frac{\|x_t^i - y\|^2}{\epsilon}\right) \mathrm{d}y = mC(\epsilon\pi)^{d(t)/2}, \tag{8}$$

where $C > 0$ is a constant that depends on how densely the points in $X_t$ populate $\mathbb{X}_t$ and $d(t)$ is the dimension of the manifold $\mathbb{X}_t$. To better understand the constants $C$ and $d(t)$, assume that $X_t$ is a large $d$–dimensional grid of gridlength $\ell$ and consider the Euclidean distance $\mathrm{dist}(x,y) = \|x - y\|$. Then, the sum $S_t(\epsilon)$ corresponds to a Riemannian integral approximation and the approximation in Eq. (8) is valid with $C$ being the average number of points per unit of volume, which is given by $C \approx \ell^{-d}$, respectively $\ell = C^{-1/d}$.

Taking the logarithm on both sides of Eq. (8) reveals an affine linear connection between $\log(S_t(\epsilon))$ and $\log(\epsilon)$:

$$\log(S_t(\epsilon)) \approx \frac{d(t)}{2}\log(\epsilon) + \log(C) + \log(m) + \frac{d(t)}{2}\log(\pi). \tag{9}$$

Hence, in order to determine a range of suitable values for $\epsilon$, we plot the function $S_t(\epsilon)$ versus $\epsilon$ in a log-log plot for each time $t$ and look for a range of $\epsilon$ in which the graph is linear. See Fig. 3 for an example. Additionally, we can read off the dimension $d(t)$, as well as a measure of density $\ell(t) = C^{-1/d(t)}$ of the point cloud $X_t$.

$$d(t) := 2 \cdot \max_{\epsilon > 0} \frac{\partial \log(S_t(\epsilon))}{\partial \log(\epsilon)} \tag{10}$$

$$\ell(t) := C^{-1/d(t)} = \left(\frac{S_t \epsilon^*}{m}\right)^{1/d(t)} (\epsilon^*\pi)^{1/2}, \tag{11}$$

where $\epsilon^* > 0$ is the value that maximizes the slope $\partial \log(S_t(\epsilon))/\partial \log(\epsilon)$. We note that the derivations of $d(t)$ and $\ell(t)$ are not mathematically rigorous and are used heuristically. In particular, the dimension $d(t)$ does not have to be an integer and $\ell(t)$ is just an approximate measure of (inverse) density when the point cloud $X_t$ is not a perfect grid. We stress that, since $\ell(t)$

approximates a grid length, higher values correspond to lower point densities. In addition, we note that $d(t)$ is invariant under isotropic contraction/expansion of $X_t$ (it is scale-invariant), but $\ell(t)$ is not.

## 2.5   Boundary handling with $\alpha$-shapes

The estimation of coherent sets requires proper handling of boundary points $\partial X_t \subset X_t$. Hence, a method is needed to determine which points lie on the boundary of a given point cloud $X_t$. This problem reduces to the well-researched problem of surface

reconstruction from point cloud data. The simplest approach is to use the uniquely defined convex hull of $X_t$. This method is too coarse for our purpose, since the point cloud $X_t$ is in general not a convex object. Once concavity is permitted, detection of a bounding surface of a set of points is ambiguous and several algorithmic approaches exist (Berger et al., 2017). We have decided on using the established `alpha shape` algorithm first introduced by Edelsbrunner et al. (1983) (cf. Edelsbrunner (2012) for an overview), since it does not require surface normals and can be conveniently tuned by only one parameter $\alpha \geq 0$.





Large values of $\alpha$ correspond to a structured surface while small values of $\alpha$ result in a smooth surface. For a thorough derivation and details on algorithmic implementation, consult Edelsbrunner and Mücke (1994).

Let $X_t \subset \mathbb{R}^n$ be a finite set of points, $0 \leq \alpha < \infty$ a real number. We denote open balls in $\mathbb{R}^n$ of radius $\alpha$ by $b_\alpha$. An $\alpha$-ball $b_\alpha$ is said to be empty, if it does not contain any points from $X_t$. The $\alpha$-hull $\mathbb{H}_\alpha$ is then defined as the complement of the union of all empty $\alpha$-balls:

$$\mathbb{H}_\alpha := \mathbb{R}^n \setminus \bigcup_{b_\alpha \cap X_t = \emptyset} b_\alpha. \tag{12}$$

We define the boundary of $X_t$ to consist of those points that lie in the boundary of $\mathbb{H}_\alpha$, i.e.

$$\partial X_t := \{x \in X_t \mid x \in \partial \mathbb{H}_\alpha\}.$$

An equivalent definition of the boundary is that $x \in X_t$ lies in $\partial X_t$ if and only if there is an empty open $\alpha$-ball $b_\alpha$ such that $x \in \partial b_\alpha$. As $\alpha \to \infty$ the set $\mathbb{H}_\alpha$ recovers the convex hull, whereas $\alpha = 0$ results in $\mathbb{H}_\alpha = X_t$. Hence, the set $\partial X_t$ grows as

$\alpha \to 0$, and for $\alpha$ small enough, we find $\partial X_t = X_t$.

This sparks the question of choosing $\alpha$ appropriately such that $\partial X_t$ defines a boundary of the point-cloud $X_t$ of the desired coarseness. As discussed in Sect. 2.4, $\sqrt{\epsilon^*}$ provides a measure of the typical distance between points in the point cloud. Therefore, it is natural to choose $\alpha$ in the same magnitude as $\sqrt{\epsilon^*}$.

The `alpha shape` algorithm assumes the points live in Euclidean space. However, in Sect. 4 we apply our methods to

atmospheric data. The atmosphere, being approximated by a spherical shell, is non-isotropic and globally non-Euclidean. Thus, constructing a hull in a Cartesian coordinate system, e.g. centered in Earth's core, is destined to fail as the large difference in scales between vertical and horizontal coordinates leads to points being sampled from a nearly two-dimensional region in space. In such a perspective, virtually all points would be boundary points. We have therefore decided to apply a stereographic projection centered at the North Pole with an undistorted latitude of $50°$ N to the horizontal coordinates and applied a linear

vertical scaling in accordance with the custom distance metric introduced in Sect. 3.4 before applying the `alpha shape` algorithm. A stereographic projection seems apt since the air parcels of our examples stay on the Northern Hemisphere and are gathered around the mid-latitudes for most of the time. Other suitable coordinate transformations alter the selected boundary points only to a small degree and do not change the resulting coherent sets detected significantly (not shown).

## 3 Implementation

All atmospheric fields used in the analysis are taken from the ERA5 reanalysis product provided by the European Centre for Medium-Range Weather Forecasts (ECMWF) (Hersbach et al., 2020). It resolves the global atmosphere on a grid with a roughly 31 km horizontal spacing and 137 hybrid vertical levels between the surface and 1 hPa on hourly timescale.



## 3.1 Blocking identification

The atmospheric blocking regions are defined as in Pfahl and Wernli (2012), who utilized the algorithm introduced by Schwierz
et al. (2004). More specifically, grid points are identified as blocked, if a vertically averaged (between 500 hPa and 150 hPa)
negative PV anomaly (with respect to the monthly climatology) larger than 1.3 potential vorticity units ($10^{-6}$ K m$^2$ kg$^{-1}$ s$^{-1}$)
(pvu) for at least five days is observed for a potentially traveling connected region (individual grid points need not experience
this anomaly for five full days). Data are available in six-hourly time steps, and PV anomalies are required to have a spatial
overlap of at least 70% to be assigned to the same track.

Though a host of different blocking identification mechanisms are available (Pinheiro et al., 2019), this PV-based algorithm
has been chosen since it encapsulates the most important dynamical features of atmospheric blocking (Schwierz et al., 2004;
Croci-Maspoli et al., 2007) and directly identifies two-dimensional regions. The identified regions will usually mark the areas
responsible dynamically for the blocking characteristics (i.e. the high-pressure ridge regions) rather than the areas marked by a
geopotential height reversal. The focus of this study lies in the demonstration of the methodological framework when applied
to atmospheric blocking air streams rather than in arriving at definitive or quantative insights about blocking consistent across
different blocking definitions, which is why we abstain from comparing results between different blocking indices.

## 3.2 Initial points

The method for the identification of coherent air streams described in Sect. 2 is applied to two case studies. Given the two–
dimensional, global, boolean field of whether a grid point is blocked or not, for each case, an atmospheric blocking event is
identified as a connected region of `True` values developing in time. For an individual time step, a respective region is "filled"
with trajectory initial points with a vertical distance of 7 hPa between 550 and 150 hPa (we choose 550 hPa instead of 500
hPa for a slightly broader scope; the same does not apply for the upper limit, since it would likely cross the tropopause) and
a horizontal distance given by the scale difference parameter $\kappa$ times the vertical distance (cf. Sect. 3.4). In our case study,
for a scaling parameter of $\kappa = 15$ km hPa$^{-1}$, a horizontal distance of 105 km was used. Such a point density has emerged as
the highest possible point density that still allowed for an acceptable computational complexity. A zero-mean Gaussian noise
with standard deviation equal to a quarter of the distance in the respective direction is then added to each point individually to
prevent the regular structure of the initial point distribution to bias the coherent set clustering later on. Finally, all initial points
that lie above the dynamic tropopause – defined as the 2 pvu isosurface – are removed.

## 3.3 Trajectory calculation

The initial points are then used to calculate three-day forward and backward trajectories from three-dimensional wind fields
on model levels using the LAGrangian ANalysis TOol (`LAGRANTO`) (Sprenger and Wernli, 2015), which employs an iterative
predictor-corrector procedure. We think of the resulting trajectories as solutions of the dynamical system describing the motion
of air parcels in the atmosphere ($x_t^i$ from Sect. 2). Various dynamical variables can be traced along the path of the air parcels'



trajectories including potential temperature $\theta$, specific humidity $q$, temperature $T$ and all coordinates and velocities. These will
reveal the dynamical properties present in the clusters generated from purely geometric information.

### 3.4  Distance Calculation

Our methodology presented in Sect. 2 is based on point-wise distance calculation. Here, again, the issue of vastly different length scales in the horizontal and vertical directions arises. Resorting to a map projection as with the `alpha shapes` algorithm described in Sect. 2.5 does, however, not seem to be the best option since errors introduced during the stereographic
projection can be avoided. This is because calculating distances between points does not necessarily require the points to live in Euclidean space. In contrast to Banisch and Koltai (2017), who relied on the Euclidean norm as a measure of distance, we therefore construct a non–Euclidean distance which connects the vertical and horizontal scales through a scale parameter $\kappa$:

$$\kappa = \frac{\overline{\sqrt{u^2+v^2}}}{\overline{|\omega|}} = \frac{\sum_{i=1}^{m}\sum_{t=0}^{T}\sqrt{(u_t^i)^2+(v_t^i)^2}}{\sum_{i=1}^{m}\sum_{t=0}^{T}|\omega_t^i|}, \tag{13}$$

where $u, v$ are the two horizontal and $\omega$ the vertical velocity component. For two points $x^i = (\varphi^i, \lambda^i, p^i)$ and $x^j = (\varphi^j, \lambda^j, p^j)$,
given by their respective latitudes $\varphi$, longitudes $\lambda$, and pressure level $p$, we then define

$$\text{dist}(x^i, x^j) = \sqrt{\text{dist}_{\text{h}}(x^i, x^j)^2 + (\kappa(p^j - p^i))^2},$$

$$\text{dist}_{\text{h}}(x^i, x^j) = 2r_{\text{E}}\arcsin\left(\sqrt{\sin^2\left(\frac{\varphi^j - \varphi^i}{2}\right) + \cos\varphi^i \cdot \cos\varphi^j \cdot \sin^2\left(\frac{\lambda^j - \lambda^i}{2}\right)}\right),$$

where $r_{\text{E}}$ stands for Earth's radius. The distance is symmetric and positive. The horizontal distance $\text{dist}_{\text{h}}$ is the Haversine distance which approximates the great-circle distance of two points on Earth's surface (assuming a spherical shape) well
(Green, 1977).

For the scale parameter $\kappa$, a heuristic approach has been chosen that estimates the difference in scales by comparing average horizontal and vertical velocities, where the average is taken over all trajectories and time steps. We think of distances to be similar if air parcels take roughly the same amount of time to overcome them given some average velocity which is the reasoning behind the construction of $\kappa$. In addition, atmospheric turbulence – the dominant source of diffusion at the scales
investigated here – roughly scales with velocity. The developed notion of distance relies only on geometric information (if one conceives velocities as geometric), which allows comparison of purely geometric coherence to similarity of dynamic properties. For the construction of $\kappa$ we have deliberately abstained from including measures of vertical stability or asymptotic methods since the phenomena investigated feature relevant processes on synoptic as well as mesoscale which would further complicate the choice of scale-connecting characteristic quantities (Klein, 2010). Note that, due to definition of $\kappa$ in Eq. (13), the custom
metric $\text{dist}(\cdot, \cdot)$ is formally measured in kilometers.

We remark that estimated values of $\kappa$ varied only by roughly 20 % and results were rather insignificant to the exact value of $\kappa$ applied in both the algorithm and the initial point generation. For ease of computation and comparability between cases, we have therefore decided to choose a constant $\kappa = 15$ km hPa$^{-1}$ across all cases investigated. Note that requiring exact equality




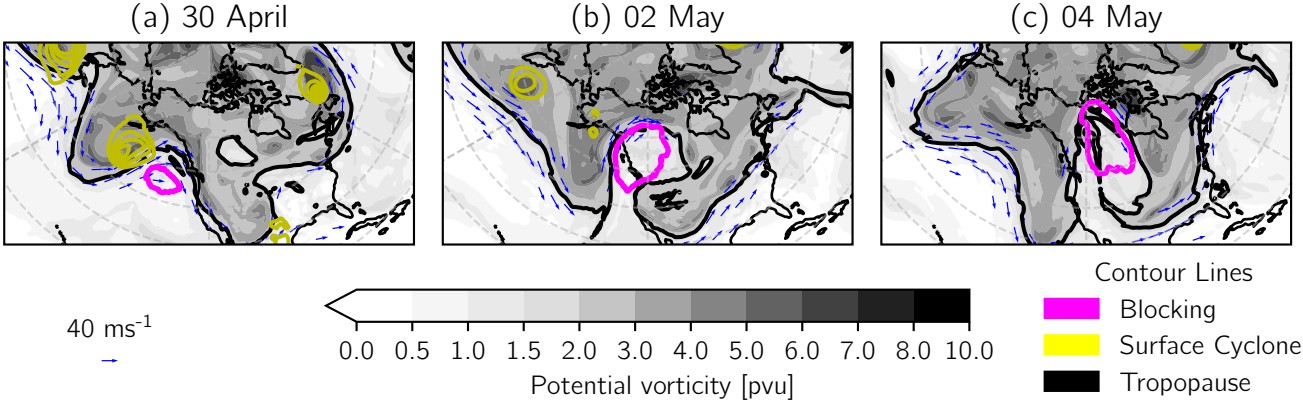

**Figure 2.** Synoptic conditions at three time instances (all at 00 UTC) during the 2016 Canada case. Upper-level PV (shaded), 2 pvu contour (tropopause) in black, upper-level wind (blue vectors, only shown for wind speeds larger than 30 ms⁻¹), surface cyclones (sea level pressure; yellow contours from 980 hPa to 1000 hPa every 5 hPa), and blocking region (magenta contour; cf. Sect. 3.1 for definition). Upper-level fields are vertically averaged between 500 and 150 hPa. Note that horizontal wind velocity is indicated by the quivers' length. Denser quivers towards the poles result from denser longitudes.

between the empirical $\kappa$ and the $\kappa$ used in the initial point generation would require extensive iteration, since the empirical $\kappa$
depends on the trajectories, which, in turn, depend on the initial point locations.

## 4 Case studies

### 4.1 Canada 2016

As a first example, the strong Ω-blocking observed from late April to early May 2016 over Canada is investigated. It is considered one of the main causes of the 2016 Fort McMurray wildfire, the costliest disaster in Canadian history up to that date (Statistics Canada, 2017). The blocking identification data set introduced in Sect. 3 identifies blocked grid points from 29 April 2016 06 UTC to 05 May 2016 00 UTC. Synoptic conditions along with the blocking region are shown for three time steps representative of onset, maintenance and decay in Fig. 2. A video of synoptic conditions at all time steps is provided in the supplement.

During the onset phase (a), the co-located surface cyclone and upper-tropospheric trough just east of the dateline induced a poleward transport of warm, low-PV air masses leading to the build-up of the strong blocking anticylcone. The pattern agreed roughly with the positive phase of the Pacific-North American (PNA) pattern. Steinfeld (2019) showed that latent heat release in air parcels transported by the surface cyclone's WCB played a significant role in formation and maintenance of the blocking. The onset was characterized by anticyclonic Rossby wave breaking to its western flank, amplifying the Eastern Pacific ridge,





which is atypical according to Miller and Wang (2022), who identified cyclonic Rossby wave breaking as the prototypical onset

mechanism for Pacific blocking (though they only investigated winter blocking).

Having acquired its large spatial extent over Western Canada (b), the blocking exhibited its prominent $\Omega$-shape with two high-PV regions to its southwesterly and southeasterly sides. The configuration severely disturbed the zonal circumpolar bands of high wind speeds (jet stream) in its eponymous blocking behavior. The following days were characterized by a low eastward propagation speed and quasi-stationary flow conditions. The blocking eventually dissolved (c) accompanied by an emergence

of a surface cyclone on its western flank (not shown).

In agreement with a magnitude of the scale parameter of $\kappa = 15$ km hPa$^{-1}$, a horizontal distance of 105 km is applied for the initial point generation. After stratospheric point removal, the number of initial points varies from 648 to 13,661 according to the size of the identified blocking region. The measured scale parameter varies between $12.60$ and $18.51$ km hPa$^{-1}$, a departure from the assumed value of 15 km hPa$^{-1}$ that is deemed insignificant and unlikely to alter the results (cf. Fig. A1 in App. A). In

fact, calculating $\kappa$ individually for every set of trajectories has been tested and did not alter the results considerably. Apart from the first few initialization dates, which feature only few trajectories, the temporal development of $\kappa$ indicates larger horizontal than vertical motion in trajectories passing through the blocking earlier, but qualitative interpretations are hard to formulate given the complex spatio-temporal dependence of $\kappa$. Mean horizontal and vertical velocities over the six–day length of the trajectories for sets of trajectories initialized at any of the dates are provided in the supplement as Fig. S1.

### 4.1.1 Trajectory density

The atmosphere being a turbulent, chaotic system, we generally expect trajectories to disperse approximately symmetrically forward and backward away from a initialization configuration ($t = 0$ h; $t$ are the hours away from the initialization time). Individual cases will, however, exhibit particularities and, more specifically, asymmetric evolution of $d(t)$ and $\ell(t)$, which measure the dimension and density of the point cloud and thus quantify the general coherence and dispersion of the air parcels

(cf. Sect. 2.4).

Figure 3 gives an example where this is the case. At any point in time, the sum of diffusion similarities $S_t(\epsilon)$ varies from $m$ to $m^2$. For low diffusion strengths $\epsilon$, the diffusion similarity matrix $\mathbf{K}_{t,\epsilon}$ is only populated on the main diagonal (approximates the identity matrix), whereas for high values, the matrix has ones everywhere. The approximately polynomial (cf. Eq. (8)) intermediate regime appearing linear on a log-log plot, however, is described by two degrees of freedom, the slope $d(t)/2$ and

offset $\ell(t)$, which were shown to be heuristically connected to the dimension and inverse density of the point cloud $\{x_t^i\}$.

In the example presented here, the curve for the initialization time $t = 0$ h has the highest slope, which is because points are placed in a regular three-dimensional grid (cf. Sect. 3.2). With increasing $|t|$ in both positive and negative direction, slopes reduce similarly, but curves for positive $t$ saturate earlier. Therefore, the point cloud traced by forward trajectories ($t > 0$ h) stays more densely packed than the same point cloud traced by backward trajectories ($t < 0$ h). Hence, while $d(72\text{h}) \approx d(-72\text{h})$,

$\ell(72\text{h}) < \ell(-72\text{h})$ (recall that $\ell(t)$ approximates a grid length and is therefore inversely related to density).

The inset illustrates, what is happening: Not only do the air parcels diffuse more rapidly horizontally before entering the blocking region compared to after, they also tend to spread to a larger degree in the vertical. A larger spread is related to a

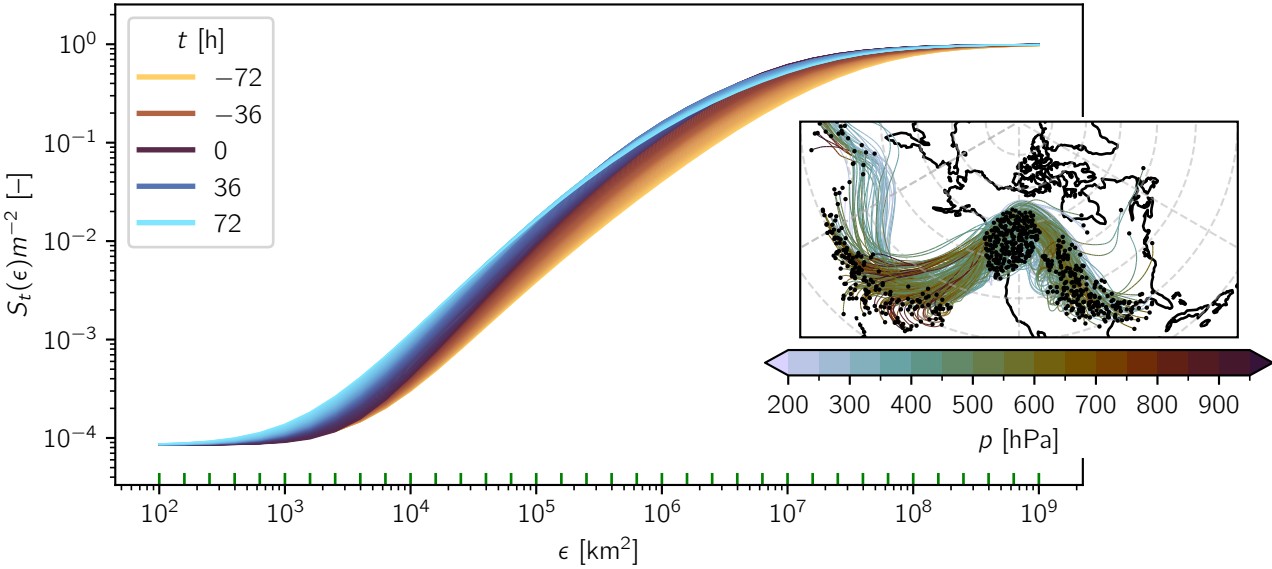

**Figure 3.** Normalized sum of pointwise diffusion similarities $S_t(\epsilon)m^{-2}$ plotted versus diffusion bandwidth $\epsilon$ on log-log scale for trajectories initialized at 02 May 2016 00 UTC. Green horizontal axis ticks indicate evaluated values of $\epsilon$. Inset: Trajectories in stereographic projection with vertical coordinate ($p$) colorcoded. Parcel locations at $t = -72\text{h}, 0\text{h}, 72\text{h}$ as black dots. Only every $50^{\text{th}}$ trajectory is shown for clarity.

smaller density of the points in space and thus larger $\ell$. The slope of individual curves is hard to determine from Fig. 3, which is why both $d(t)$ and $\ell(t)$ are displayed for the whole life cycle of the blocking and the according three–day forward and backward trajectories in Fig. 4. The data is ordered according to the initialization time along the vertical axis and according to the trajectories' time steps along the horizontal axis. Since trajectories are initialized every six hours and trajectory data is hourly, white lines with a slope of $1/6$ indicated isochrones. Data belonging to the initialization date 29 April 2016 06 UTC has been omitted due to the low number of trajectories.

The dimension heuristic $d(t)$ is higher the closer the air parcels are in time to the initialization configuration ($t = 0$ h). Even then, however, the theoretically achievable dimension of 3 is not reached, which is a boundary artifact, since for points near the boundary, the Gaussian function in Eq. (8) is not fully resolved. This is also why a higher number of trajectories leads to higher values of $d(t)$. Furthermore, Fig. 4 (a) bears testimony of the fact that points tend to arrange more two-dimensionally the further they get away from the initialization time, though two distinct regions may be identified, where this is not the case.

Trajectories passing through the blocking during its late maintenance phase ($\approx$ 03 May; lower center left region in the plot) exhibit a higher dimensionality during their journey into the blocking ($t < 0$) and trajectories passing through the region during its early maintenance phase ($\approx$ 02 May; upper center right region in the plot) exhibit an increased dimensionality during their journey out of the region ($t > 0$). The size of the blocking is of first order importance for this phenomenon, since high pressure



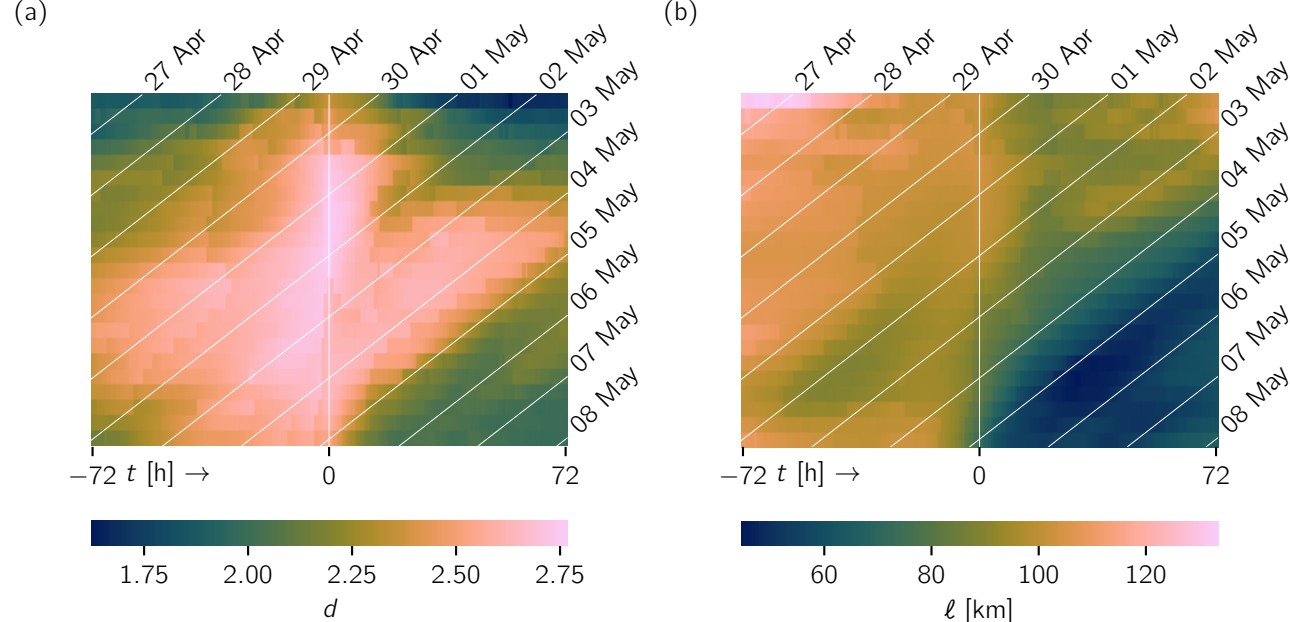

**Figure 4.** (a) $d(t)$ and (b) $\ell(t)$ plotted as a function of initialization time (23 initializations, vertical axis) and time step (144 time steps from $-72$ h to $+72$ h, horizontal axis) for the Canada 2016 case. Data corresponding to trajectories initialized at 29 April 2016 06 UTC has been removed. White diagonals indicate isochrones (all 00 UTC). Vertical white line indicates initialization time.

regions suppress both horizontal and vertical motion (at least in their centers). The larger the blocking, the larger the fraction of air parcels that are under its influence.

Inspection of the individual trajectories confirms that the quasi-stationary flow field reduces the amount of filamentation the point cloud is subjected to. This is because both spatial and temporal small-scale variability will shear – and thus filament – a spatially extended point cloud. In contrast, the air masses traced here, behave more like a rigid body being translated and rotated by the lateral large scale jet and the flow inside the blocking. Recall that $d(t)$ is scale-invariant and isotropic contraction/expansion will, therefore, not change $d(t)$ (cf. Sect. 2.4).

The level sets of $d$ coinciding with the isochrones for $2.4 > d > 2.2$ at the bottom right of panel (a) indicate several traced air masses that pass through the blocking from the 02 May onwards experience a considerable decrease in $d$ roughly at the same time $\approx 05$ May 00 UTC. The associated straining motion is related to the disintegration of the blocking and its accompanying lateral wind bands followed by the reestablishment of the zonal jet. The reorganization of the large–scale flow field exposes the air masses to considerable shear, effectively filamenting the individual point clouds and reducing $d(t)$.

The density measure $\ell(t)$ shown in Fig. 4 (b) sheds more light on the spatial nature of the investigated trajectories. Importantly, $\ell(t)$ does not measure the general density of the point clouds (cf. Fig. A2 in App. A for mean nearest neighbor point distances), but rather the inverse density of the points $X_t$ populating $\mathbb{X}_t$, respecting its dimension. This explains, why $\ell(t)$ does





not assume its lowest values uniformly at $t \approx 0$, where values equate to roughly 105 km in agreement with the initial point generation.

The approximated grid length $\ell(t)$ tends to be lower for forward trajectories compared to backward trajectories. This is especially true for trajectories initialized during the second half of the blocking's life cycle. We find coincidence of level sets of $\ell$ with isochrones between 01 and 02 May in the bottom left of the plot and around the 05 May on the botton right of the plot. In both cases, $\ell$ decreases with $t$, in the first instance with only a slight change in $d$ and in the second instance with a concurrent decrease in $d$. Hence, the first case hints at isotropic contraction of the point cloud, which is caused by the traced air parcels'

arrival in the blocking. For the second instance, the straining imposed on the point cloud of air parcels by the reorganizing flow field makes $\mathbb{X}_t$ appear more two-dimensionally without (sufficient) concurrent isotropic expansion, such that $\ell(t)$ decreases.

     All in all, evolution of the air parcel point clouds traced by the trajectories can be differentiated with respect to the blocking's lifecycle: Parcels passing through the blocking during onset experience filamentation and straining in both directions in time away from the initialization time ($t = 0$), but are a little more densely packed after leaving the blocking ($t > 0$), which is

partially due to a smaller vertical extent. Parcels passing through the blocking during its maintenance phase experience less straining, especially once inside the blocking, but become more densely packed. And parcels passing through the blocking during its decay experience strong and abrupt filamentation upon the blocking's disintegration, which coincides with a decrease in both $d$ and $\ell$.

     The filamentation of $\mathbb{X}_t$ can also be understood from a more theoretical perspective. Under the influence of the disintegrating

blocking, the air masses exhibiting a rapid decrease of $d$ and $\ell$ undergo material elongation due to the well-researched enstrophy cascade present in turbulent systems with two-dimensional configuration, which is an approximation of the atmosphere due to the scale difference between horizontal and vertical motion (Ditlevsen, 2010; Boffetta and Ecke, 2011). Such an enstrophy cascade to smaller scales and an energy cascade to larger scales involves the elongation and eventual scattering of vortical structures, which explains a decrease of dimension towards $d = 2$. The reconfiguration of the jet stream from a wavy shape

(energy at high wavenumber) to a more zonal one (energy at low wavenumber) supports this hypothesis.

     The vortex scattering represents a break up of stability aggregated during the blocking onset and maintenance. The quasi-stationarity of trajectories visible mainly in the maintenance phase of the blocking has to come to an end eventually, already from an entropy-perspective. In a structureless, random and turbulent fluid dynamical system, one would expect a more or less symmetric distribution with respect to distance from $t = 0$ in $d(t)$ (cf. Fig. 4 (a)). Thus, the stability attributed to blocking

mainly with respect to flow configuration seems to be applicable also to the material air parcels passing through the region.

### 4.1.2 Coherent Air Streams

In order to objectively identify WCB air streams stabilizing the blocking as hypothesized by Steinfeld and Pfahl (2019), we construct the averaged diffusion operator $\mathbf{Q}_\epsilon$ using $m = 11,774$ backward trajectories initialized in the blocking region at 02 May 2016 00 UTC ($-72\,\mathrm{h} \leq t \leq 0\,\mathrm{h}$) for a range of values of $\epsilon$ between $2 \times 10^4\,\mathrm{km}^2$ and $5 \times 10^5\,\mathrm{km}^2$; the range for which the

curve of $S_t(\epsilon)$ over $\epsilon$ appears linear in a log-log plot (cf. Fig. 3). For boundary point detection, we apply a value of $\alpha = 10^3$ km,





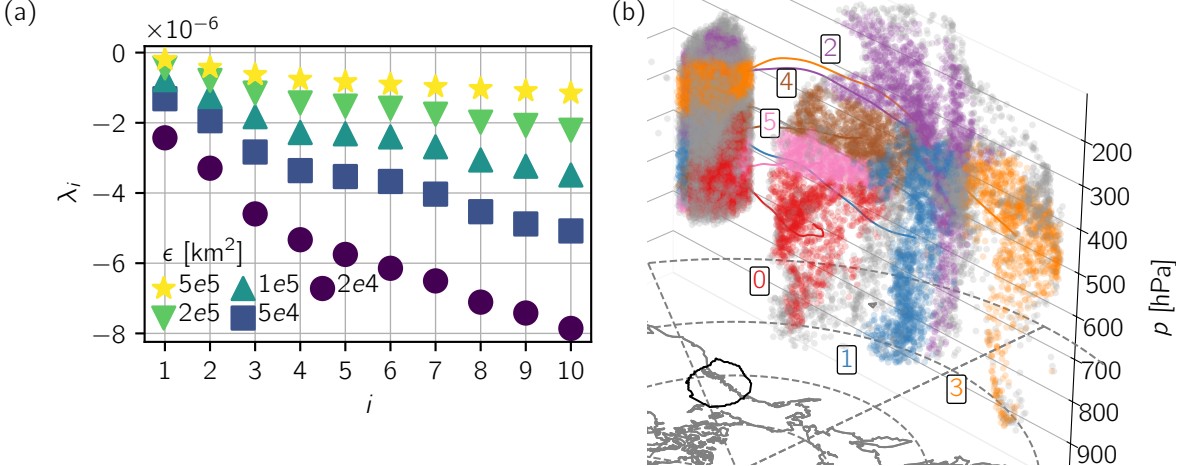

**Figure 5.** Coherent sets for backward trajectories initialized at 02 May 2016 00UTC. (a): 10 largest eigenvalues of $\mathbf{L}_\epsilon = (\mathbf{Q}_\epsilon - \mathbf{I})/\epsilon$ for various values of $\epsilon$. (b): Clustered trajectories for $\epsilon = 5 \times 10^4$ km²: colors and according numbers show the clustering, points are shown for $t = 0$ h and $t = -72$ h, lines show average trajectories. Horizontal coordinates are in stereographic projection. Black contour shows location of identified blocking at $t = 0$ h.

which is on the conservative (higher) side of $\sqrt{\epsilon}$ for the given range of values of $\epsilon$ and find that the edge lengths of the resulting simplicial complexes at any $t$ stay between $10^2$ km and $10^3$ km (cf. Fig. A3 in App. A).

The resulting spectrum of $\mathbf{L}_\epsilon = (\mathbf{Q}_\epsilon - \mathbf{I})/\epsilon$ is shown in Fig. 5 (a). We show the spectrum of $\mathbf{L}_\epsilon$ instead of $\mathbf{Q}_\epsilon$ because $\epsilon$ controls how close $\mathbf{Q}_\epsilon$ is to the identity matrix (cf. Fig. 3) and, hence, how close the eigenvalues are to one. The largest
eigenvalues are close but not equal to zero, which is caused by the application of boundary conditions to the normalized (stochastic) matrices $\mathbf{P}_{\epsilon,t}$ before averaging them to obtain $\mathbf{Q}_\epsilon$. We identify a moderate spectral gap after the seventh eigenvalue (disregarding spectral gaps at $i = 2, 3$ in order to achieve sufficient detail) and perform `k-means` clustering of the data points into seven clusters in the coordinate space spanned by the first six eigenvectors using a value of $\epsilon = 5 \times 10^4$ km². We found that the resulting clusters are remarkably robust to variation of $\epsilon$ and the number of clusters.

Figure 5 (b) depicts the resulting clusters differentiated by colors. Shown are locations of the air parcels at both the initialization ($t = 0$ h) and the end point ($t = -72$ h) as well as each cluster's average trajectory (calculated by all points' average location for each time step). The gray cluster represents the set of points for which all eigenvectors of $\mathbf{Q}_\epsilon$ simultaneously show values close to zero. This implies that the points in the gray cluster are, most of the time, boundary points. The remaining six clusters show remarkable coherence upon visual inspection of the points at each $t$, though the coherence tends to be stronger
the closer the parcels are to the initialization, which seems natural as the points more strongly resemble a three-dimensional continuum the further they have traveled towards their initialization point in the blocking.

Figure 6 (a) provides insight into the dynamical properties of the individual clusters, though the gray boundary cluster has been omitted. Importantly, we identify a considerable increase in potential temperature for the air parcels in the red (0) and blue



(1) clusters. The red cluster starts at the eastern flank of $X_{-72}$ in the lower troposphere and travels to the northern, lower flank of $X_0$ undergoing significant upward motion. In absence of other strong diabatic effects, the large positive change in potential temperature can only be caused by latent heating, which is confirmed by its change in specific humidity (cf. supplementary Fig. S3). The change occurs mainly during the 24 hours leading up to the arrival in the blocking, which is in contrast to the blue cluster, which experiences latent heating earlier. The magnitude of the heating in the blue cluster is also a little lower, since the cluster initially extents in the vertical to a larger degree. The cluster undergoes vertical compression over the course of the three days and ends up in the lower southeasterly corner of the low-PV anomaly region, adjacent to the red cluster. The diabatic lifting of both clusters is due to large scale ascent typically associated with warm fronts. The blue cluster "overtakes" the red, brown (4) and pink (5) clusters as it reaches the region of strong lower-tropospheric winds earlier and remains in the blocking region after its earlier arrival while slowly progressing under anticyclonal movement. The behavior of both clusters strongly suggest the existence of a warm conveyor air stream caused by the strong adjacent surface low (cf. Fig. 2 (a)), which supplies the blocking with anomalously low PV air masses at two distinct points in time.

Forward trajectories for the same initialization time exhibit synchronized behavior similar to the example discussed next and will thus not be discussed here. Changes in potential temperature are almost exclusively negative, hinting at radiative cooling typical for air masses in the upper troposphere. The synchronization of movement is also indicated in Fig. 4 – the air masses barely separate and are thus rather similar both geometrically and dynamically.

To understand the occurrence of WCBs across the lifetime of the blocking, we automatically identify them for each set of trajectories. To that end, we apply our algorithm with $\alpha = 10^3$ km, $\epsilon = 5 \times 10^4$ km$^2$ to backward trajectories initialized at each of the six hourly time steps. We identify the spectral gap we are interested in by finding the largest absolute difference between two sequential eigenvalues, starting from the third eigenvalue. For the trajectory clusters identified, we select those, whose median potential temperature three days before arriving in the blocking ($t = -72$ h) is at least 5 K below the median potential temperature when arriving in the blocking ($t = 0$ h). The results are shown in Fig. 6 (b). We note that this diagnostic suffers from the ambiguity of the concept of the spectral gap and, hence, the number of clusters selected, which is why we also show the clusters' sizes (number of trajectories in a cluster; $m_i$).

The strongest large scale lifting and, thus, diabatic heating occurs during the onset of the blocking, governed by the strong surface cyclone west of it. This is in accordance with findings from Miller and Wang (2022) who highlighted diabatic heating as a key mechanism for the onset of Pacific blockings (though, again, they only studied winter cases). We find coherent bundles of trajectories with median changes in potential temperature of up to 20 K within a day, which does not seem to be atypical (e.g. Madonna et al. (2014)). In that phase, the majority of trajectories are part of some cluster featuring considerable heating, as indicated by the the gray bars in the background of Fig. 6 (b). The continuous decrease in the ratio of such trajectories inside WCBs to the total number of trajectories comes about due to four different reasons. The first reason is the general enlargement of the blocking. The quasi-stationary nature of blockings implies that air parcels can get "trapped" inside it (see also the following example), which means that even parcels that have experienced latent heating at some point will not be identified as such as soon as this heating is more than three days past. The second reason is the decay of the neighboring surface cyclone,





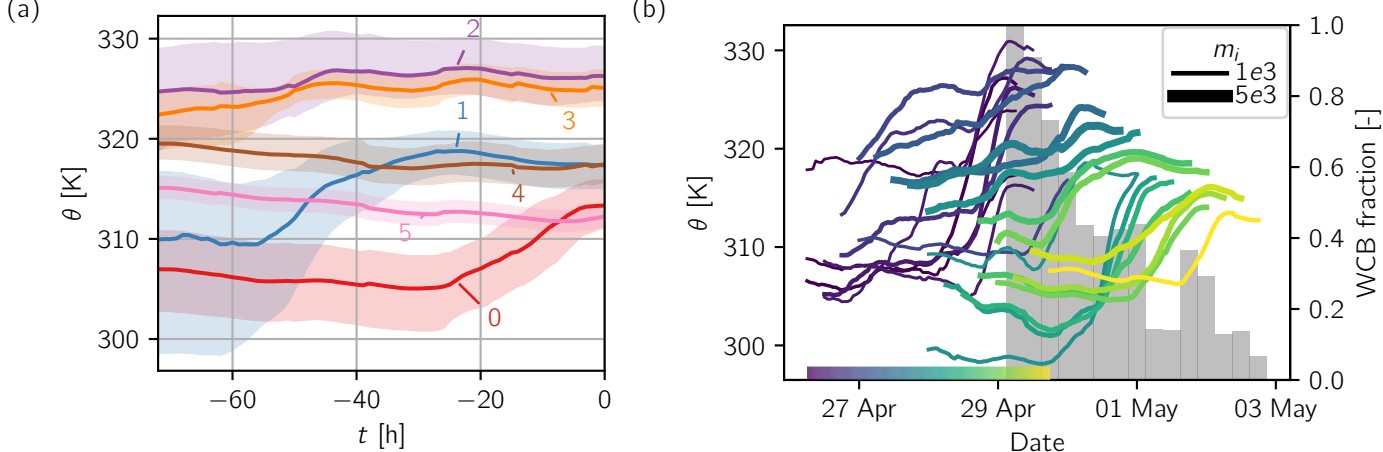

**Figure 6.** (a): Median (line) and interquartile range (shading) of potential temperature distribution among individual clusters from Fig. 5. Boundary point cluster (gray) not shown. (b): Median (line) of potential temperature distribution among clusters that exhibit a difference between initial and end value larger than 5 K for different initialization times indicated by line color. Line thickness is according to number of trajectories in cluster ($m_i$). Gray bar chart in the background shows ratio between number of trajectories contained in WCB clusters to the total number of trajectories for each arrival (initialization) time.

which is the dynamic cause of the WCBs. The third and fourth reasons are inavailability of moisture as the blocking travels over land and large scale subsidence caused by the blocking itself. Higher occurrence of WCBs during onset is confirmed by exiting research (Steinfeld and Pfahl, 2019; Hauser et al., 2023)

The plot also shows that WCB clusters identified in sets of trajectories initialized at different times tend to exhibit strong latent heating at the same time. This is because WCBs are synoptic scale structures, extensive in both space and time. Their intermittent occurrence is linked to presence and absence of surface cyclones and their corresponding warm fronts (Madonna et al., 2014; Catto et al., 2015). Note also that the general decrease in potential temperature is related to the blocking traveling poleward.

In addition to identifying WCBs, we want to further understand the behavior of $d(t)$ and $\ell(t)$ discussed in Sect. 4.1.1. To that end, we apply our algorithm to $m = 9,882$ connected forward and backward trajectories ($-72\text{h} \leq t \leq 72\text{h}$) passing through the blocking at 04 May 2016 00 UTC, which is close to the end of its lifecycle, using the same range of values for $\epsilon$ and the same value for $\alpha$ (cf. Fig. A3 in App. A for distribution of $\alpha$-hull edge lengths). We identify a moderate spectral gap after the fourth eigenvalue (cf. supplementary Fig. S4 (a)), and again apply k-means clustering using $\epsilon = 5 \times 10^4$ km$^2$, but looking for four clusters this time. The resulting clusters are shown in Fig. 7 three days before and after initialization ($t = -72$ h and $t = 72$ h) along with their mean trajectories and the blocking's location at $t = 0$ h. We do not find a cluster consisting of boundary points mostly, which is probably due to the low number of clusters and considerable mixing of points at the boundary.

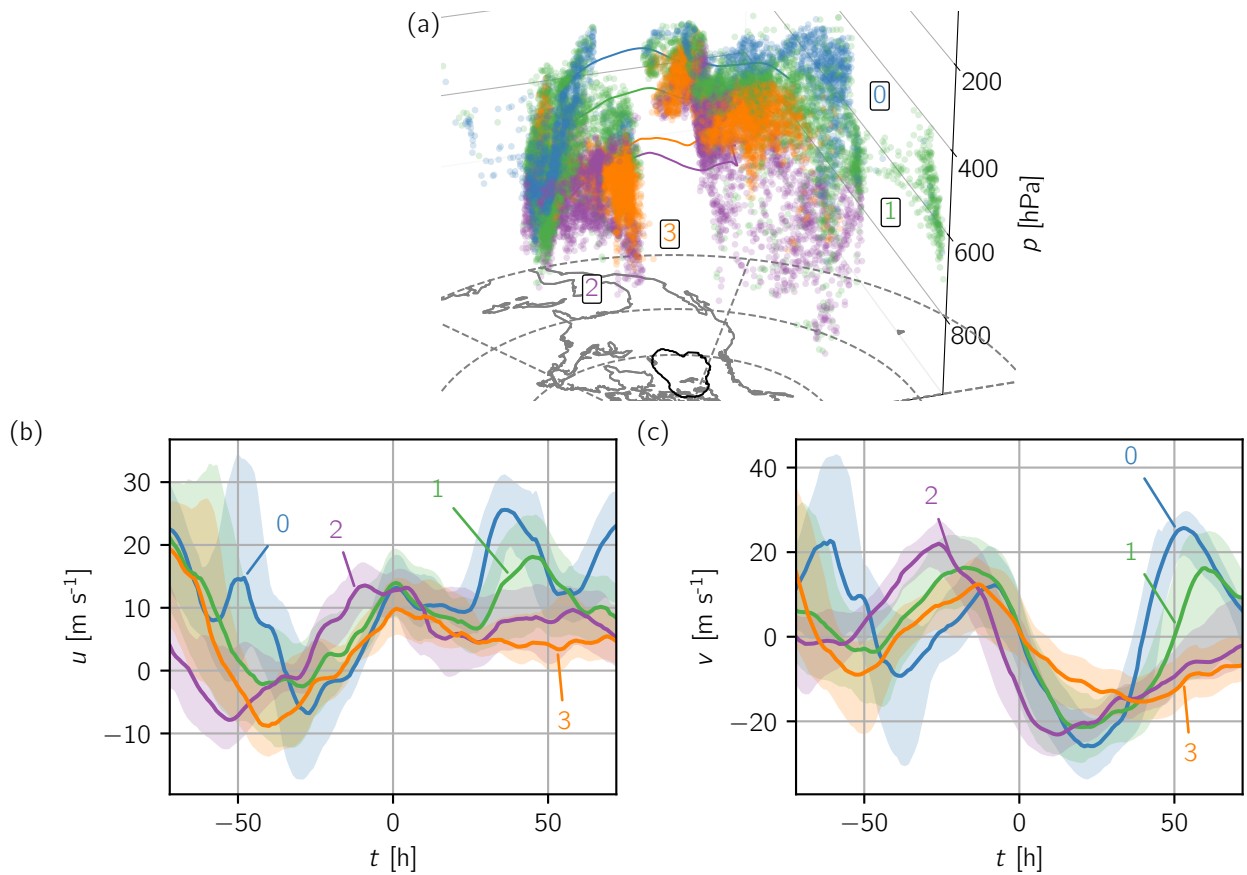

**Figure 7.** (a): Same as Fig. 5 (b) but for combined forward and backward trajectories initialized at 04 May 2016 00 UTC. Points are shown for $t = -72$ h (right bulk) and $t = 72$ h (left bulk). Black contour shows location of identified blocking at $t = 0$ h. (b) & (c): Same as Fig. 6 (a) but for horizontal velocity components. All clusters are shown.

In our experience, boundary clusters were also far more likely to occur for exclusive backward ($-72$h $< t < 0$h) or forward ($0$h $< t < 72$h) trajectories.

In contrast to the example above, the clusters are separated to a larger degree in the vertical, which is caused by predominantly synchronized movement in the horizontal as well as absence of any updrafts, as will be shown below. The air parcels move considerably less far both in the horizontal and in the vertical, especially during their approach into the blocking. In fact, the bulk of the air parcels are already within or in close vicinity to the anomaly region three days before – some are even east of the blocking. Therefore, it comes as no surprise, that latent heating does not play a role for the air parcels traced (cf. Fig. 6 (b)).

The vast majority of parcels stays in the upper troposphere between 300 and 600 hPa the whole time. While the orange (3) cluster undergoes descending motion throughout, the purple (2) cluster performs a moderate ascent into the blocking, though



all clusters descend after passage through the blocking, which is no surprise, since the initialization points populate the whole upper troposphere and passage through the tropopause is generally unlikely.

To investigate closer the movement of the parcels, Fig. 7 (b) and (c) displays the horizontal velocities of the respective clusters. Most of the air parcels experience zonal velocities below a magnitude of $10$ m s$^{-1}$ throughout their journey, but especially during their approach ($-48\text{h} < t < 0\text{h}$), which reflects the blocking's eponymous obstruction of mid-latitudinal westerlies. The obstruction also explains the larger meridional velocitiy magnitudes visible in Fig. 7 (c).

Judging from both Fig. 4 and Fig. 7, the six-day journey of the clusters (or more precisely, the air parcels) can be roughly divided into four phases:

1. Day 1 (01 May; $-72\text{h} < t < -48\text{h}$): The brown (0) and the orange (3) clusters are already in vicinity of the blocking region, while the green (1) and purple (2) clusters contain "tails" that gradually travel eastward. Those parcels already within the vicinity of the blocking are subject to the quasi-stationary anticyclonal flow field, such that $\mathbb{X}_t$ seems to curl in. The purple cluster is located on the southern flank of the blocking and, therefore, experiences westward velocities. The increase in $d$ and a decrease in $\ell$ (cf. Fig. 4) is a result of convergence horizontally and vertically and the advent of the trailing air parcels into the blocking reducing the amount of filamentation of $\mathbb{X}_t$.

2. Days 2–3 (02–03 May, $-48\text{h} < t < 0\text{h}$): All four clusters are quite close horizontally, but both the purple and the green clusters still gather some trailing parcels vertically, resulting in the continued convergence and increase in $d$. Shearing deformation is not present, rather all clusters stay within the blocking region sequentially experiencing the same velocities. In Fig. 7, both horizontal velocities show parallel curves for the different clusters, indicating a simple time shift between similar movements.

3. Day 4 (04 May, $0\text{h} < t < 24\text{h}$): The movement pattern of the different clusters continues under addition of (mostly) meridional translation as the air masses travel out of the blocking into the northerly jet on the blocking's eastern flank. Vertical motion is almost non-existent, but transport out of the blocking starts to shear $\mathbb{X}_t$, reducing $d$ and $\ell$.

4. Day 5–6 (05–06 May, $24\text{h} < t < 72\text{h}$): Under combined isentropic downward motion caused by movement out of the high pressure region, all clusters get strained horizontally along the cyclonic flow field to the southeast of the blocking's last position above the Great Lakes. Both trailing air parcels belonging to the lowest (purple) cluster and leading air parcels belonging to the highest (blue) cluster experience a strong westerly flow caused by a cyclone located above northeastern Canada. Both of these effects elongate and disperse $\mathbb{X}_t$ resulting in a rapid decrease in both $d$ and $\ell$ as the anticyclonic vortex has been scattered by neighboring cyclones working towards the reestablishment of the westerly jet stream.

## 4.2 Northern Europe 2017

The second example revolves around a blocking originating over the North Atlantic during winter, suggesting different characteristics than the case discussed above. The boreal winter 2016/2017 was marked by a number of severe cold spells over





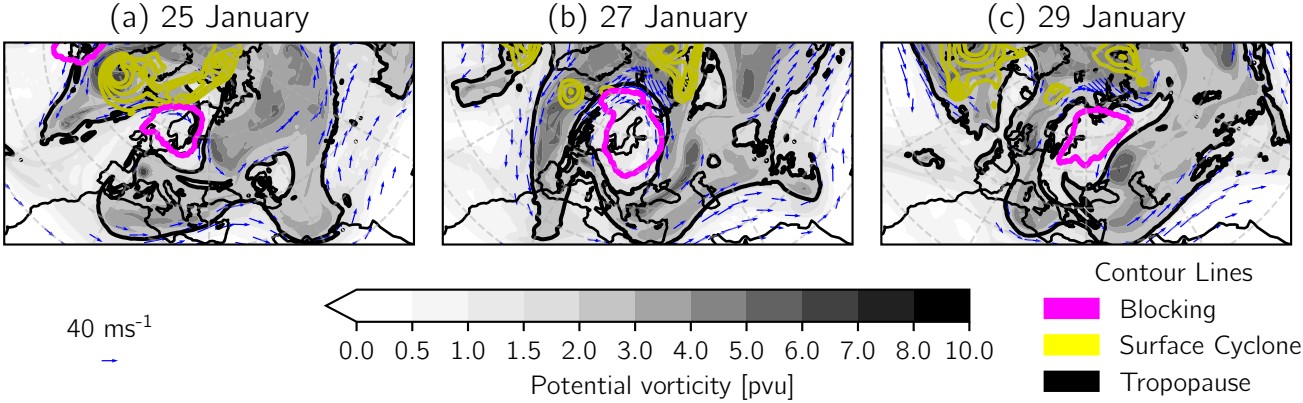

**Figure 8.** Same as Fig. 2 but for the Northern Europe 2017 case. Surface pressure intervals are from 970 hPa to 990 hPa every 5 hPa.

Central and Eastern Europe and Russia, accompanied by warm conditions in the Arctic, a low sea ice extent especially in the Barents–Kara–Sea and an exceptionally weak polar vortex. The conditions likely both favored and were enhanced by notably high blocking activity throughout mid-latitudinal Eurasia (Tyrlis et al., 2019). The conditions drastically influenced the lifes of 550 millions of people (Anagnostopoulou et al., 2017; Demirtaş, 2022; Kostopoulou, 2023).

The blocking investigated here was just the last in a number of strong European blockings, but has reached a considerable extent both spatially and temporally. This suggests low-frequency variability as a possible cause for onset as concluded by Nakamura et al. (1997), though Miller and Wang (2022) pointed to high-frequency components as key factors. Our blocking data set recognizes blocked grid points between 23 January 2017 12 UTC and 30 January 2017 12 UTC. Figure 8 shows the 555 synoptic conditions over the region of influence for three selected times representative for onset, maintenance and decay of the blocking (a video containing synoptic conditions for all time steps is provided in the supplement).

During onset (a) the low-PV anomaly developed as a vast region of low pressure over the North Atlantic and the Arctic lets subtropical air travel north, disrupting the jet stream. Blocking formation can be seen as an example of anticyclonal Rossby wave breaking, which is in agreement with Miller and Wang (2022). The blocking progressed remarkably little over the course 560 of the ensuing week and severely obstructed the usual westerly flow of air. A typical $\Omega$-blocking configuration is visible in (b) as two regions of high PV develop at the lower lateral flanks of the low-PV anomaly region. This leads to a considerable deflection of the jet stream and low-PV air masses are guided northwards on the western flank just as high-PV air masses are guided southward on the eastern flank of the blocking along strong meridional wind bands.

The blocking eventually traveled eastward, gradually dispersing over the Ural mountains and Siberia (c). Insights gained 565 from the previous case study corroborate the argument of Steinfeld and Pfahl (2019) that WCBs tend to strengthen the negative PV anomaly, which would imply that absence of which makes decay of the blocking more likely. In the present case, the blocking persisted for an extended amount of time over Scandinavia and eventually dissolved over mainland Russia, such that



a possible hypothesis is that the decay of the blocking coincided with the cessation of WCBs feeding due to a lack of moisture, a weakening of the adjacent surface cyclone and large-scale subsidence.

To analyze air stream coherence, we apply the same value for the scale parameter $\kappa = 15$ km hPa$^{-1}$ and obtain between 50 and 19,200 initial points (cf. Fig. A1 in App. A). The variation of $\kappa$ is, again, inside a moderate range of 12.09 km hPa$^{-1}$ to 19.82 km hPa$^{-1}$ and does not show a great influence on the eventual clustering of trajectories. Temporal evolution of $\kappa$ indicates relatively weaker motion in the vertical, the later the trajectories pass through the blocking. Average velocities both in horizontal and vertical direction develop similarly and largely decrease with time, which is related to the presence and absence

of WCBs and the traced air parcels' vicinity to the stagnant flow field dominated by the blocking. Mean velocities over the trajectories' whole six-day time period are provided in the supplement as Fig. S2.

### 4.2.1 Trajectory Density

We show the estimates of both $d(t)$ and $\ell(t)$ for the present example in Fig. 9. Data corresponding to trajectories initialized at 23 January 2017 12 UTC have been removed, since the initial point generation resulted in only 50 trajectories. We find that the

distributions of $d(t)$ and $\ell(t)$ over time and across the blocking's life cycle bears some resemblance to the first case, though we note that values are generally slightly higher for both heuristics, which we attribute to the higher number of trajectories. For $d(t)$, we identify maxima of well below 3 for point clouds with the most points (cf. Fig. 9) and close to the initialization time and recognize a more or less monotonic decrease away from those. Regions of larger $d(t)$ are, again, visible for $t > 0$ h for trajectories initialized during the maturing phase (25–26 January) and for $t < 0$ h for trajectories initialized during the

late maintenance phase (27–28 January). This sparks the question, whether the same physical processes are responsible for the observations as in the first case study.

Onset and early maintenance of the blocking were heavily influenced by the strong low pressure region to the west of it. Both $d$ and $\ell$ increase considerably as air is transported into the blocking along the strong wind band on the surface cyclones' eastern flank. Similar to rolling up a fish net (the rotational axis in the $p$ direction), $\mathbb{X}_t$ contracts and increases in dimension as

the parcels end up in the blocking and stay there, while the approximated grid length increases slightly. Considerable upward motion supports this process. This is in contrast to the first case study, where $\ell$ stayed largely constant during this phase. The compressed, almost three dimensional $\mathbb{X}_t$ stays largely intact and inside the blocking as it stagnates and grows over Northern Europe. Consequently, temporal development of $d$ and $\ell$ in approaching air parcels during the blocking's period of largest extent is also dominated by parcels already in or close to the blocking, such that neither $d$, nor $\ell$ change considerably.

The blocking reached its peak extent around the 27 January (cf. Fig. A1). Subsequent shrinking means air masses under the influence of the quasi-stationary flow field inside the blocking now escape and come under the influence of a low pressure center in the upper troposphere at the southeasterly flank of the blocking. This is why forward trajectories initialized around 25–26 January see essentially no decrease in $d$ or $\ell$, but those initialized around 27 January do. This is in contrast to the first example case discussed, which showed both a later peak and a slower subsequent decline in size. Disintegration of the blocking

and its associated flow field leads to decreases in both heuristics similar to the first case study, albeit less pronounced.





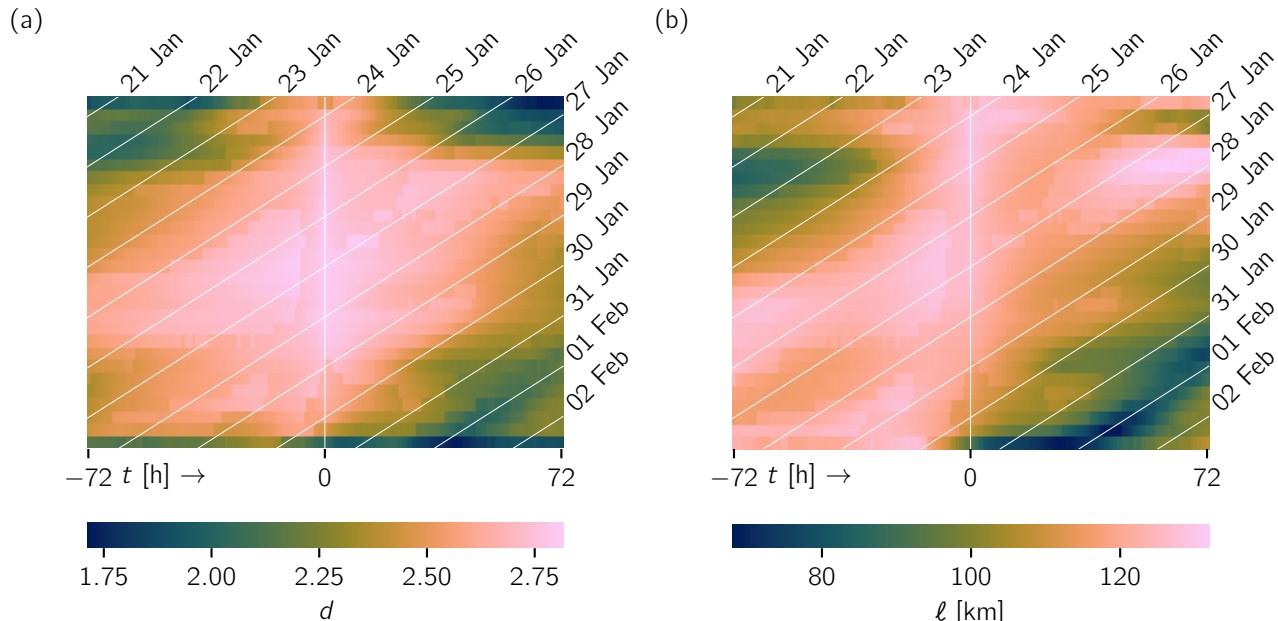

**Figure 9.** Same as Fig. 4, but for the 2017 Northern Europe case. Data corresponding to trajectories initialized at the 23 January 2017 12 UTC has been removed.

### 4.2.2 Coherent Air Streams

We apply our diagnostic presented in Fig. 6 (b) for the present case to study and summarize the occurrence of coherent air streams featuring latent heating and show results in Fig. 10 (b). WCBs are found almost exclusively among backward trajectories arriving in the blocking during its lifetime's first half. During that period, however, they make up a considerable share of all trajectories traced. Miller and Wang (2022) also identified diabatic heating as a major cause for blocking onset in the European region.

Both example cases presented here feature WCBs (according to our definition: coherent air streams with a three-day change in median potential temperature of more than 5 K) almost exclusively during the onset and early maintenance phase. In both cases, these are caused by adjacent surface lows. Compared to the first case study, the WCB fraction decreases less monotonically. A possible explanation is that moisture availability is dependant on oceanic sources to a larger degree in winter (Pfahl et al., 2014), but proving this would require moisture source tracking, which is outside the scope of the present study. In comparison to the first evaluation of WCBs, the potential temperature levels tend to be a bit lower in this case, which reflects the different seasons they occurred in. Notably, the second blocking does not show lower levels of potential temperature with time since it does not move to higher latitudes.

As a last demonstration of our methodology, we pick the set of backward trajectories initialized at 26 January 2017 18 UTC, which we understand as a turning point in the blocking's lifecycle in several senses. Firstly, at this point in time the blocking has



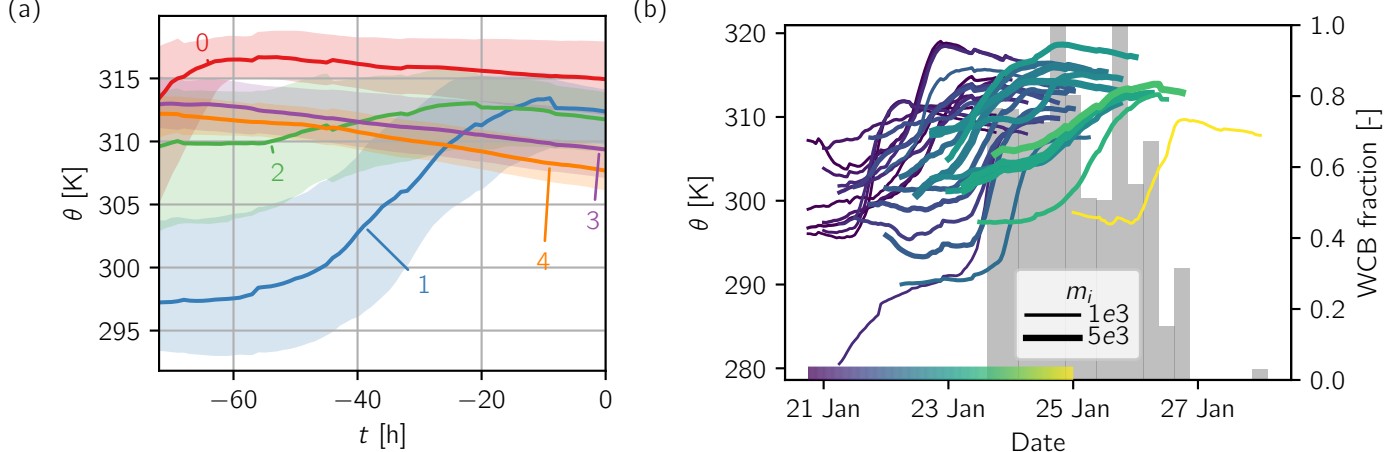

**Figure 10.** Same as Fig. 6 but for the 2017 Northern Europe case with clusters in (a) from trajectories initialized at 26 January 2017 18 UTC. In (a), a boundary cluster has been removed.

almost reached its maximum extent allowing for a total of 18,349 individual initial points. Secondly, from a process-oriented perspective, Fig. 10 (b) reveals that this point in time is the last featuring a WCB with considerable size, but the trajectories also exhibit the behavior responsible for profiles of $d$ and $\ell$ characteristic for later phases, which will be shown in the following.
The synoptic conditions at the selected point in time differ only marginally compared to Fig. 8 (b).

After calculating boundary points with $\alpha = 10^3$ km again (cf. Fig. A3 for resulting $\alpha$-hull edge length distribution), we identify a moderate spectral gap after six eigenvalues (cf. supplementary Fig. S4 (b)) and select $\epsilon = 2 \times 10^4$ km$^2$ to cluster the trajectories. The result is shown in Fig. 11 (a), where a cluster with boundary points has already been removed for visibility. Note also that the positions of the air parcels is only shown at $t = -72$ h whereas the blocking's location is shown at time $t = 0$
h. This is because air parcel locations overlap considerably for $t = -72$ h and $t = 0$ h and would therefore be impossible to distinguish.

This time, clusters are separated stronger horizontally owing to a larger degree of vertical mixing compared to the previous example. Clusters at $t = -72$ h appear rather filamented (except for the purple cluster (3)) and even feature seemingly disconnected pockets of air (blue cluster (1)), which highlights the role the final configuration close to $t = 0$ h plays. The distributions
of $\theta$ of the individual clusters over time shown in Fig. 10 (a), however, prove that all but the blue clusters are concentrated in the upper levels of the atmosphere. The blue cluster, on the other hand, can be identified as one of the WCBs visible in Fig. 10 (b). Its air parcels undergo latent heating, increasing their median potential temperature by roughly 15 K. Given the fact that the synoptic flow field and general conditions makes it likely that the other clusters have a similar "history" and have, thus, undergone a diabatic ascent as well – just earlier – further underlines the importance of latent heating in this case.
The purple cluster consists of air that is inside the blocking already at $t = -72$ h. During the ensuing three days, it performs an anticyclonic rotation characteristic of high pressure systems (in the Northern Hemisphere). This is visible in Fig. 11 (b),





(a)

(b)

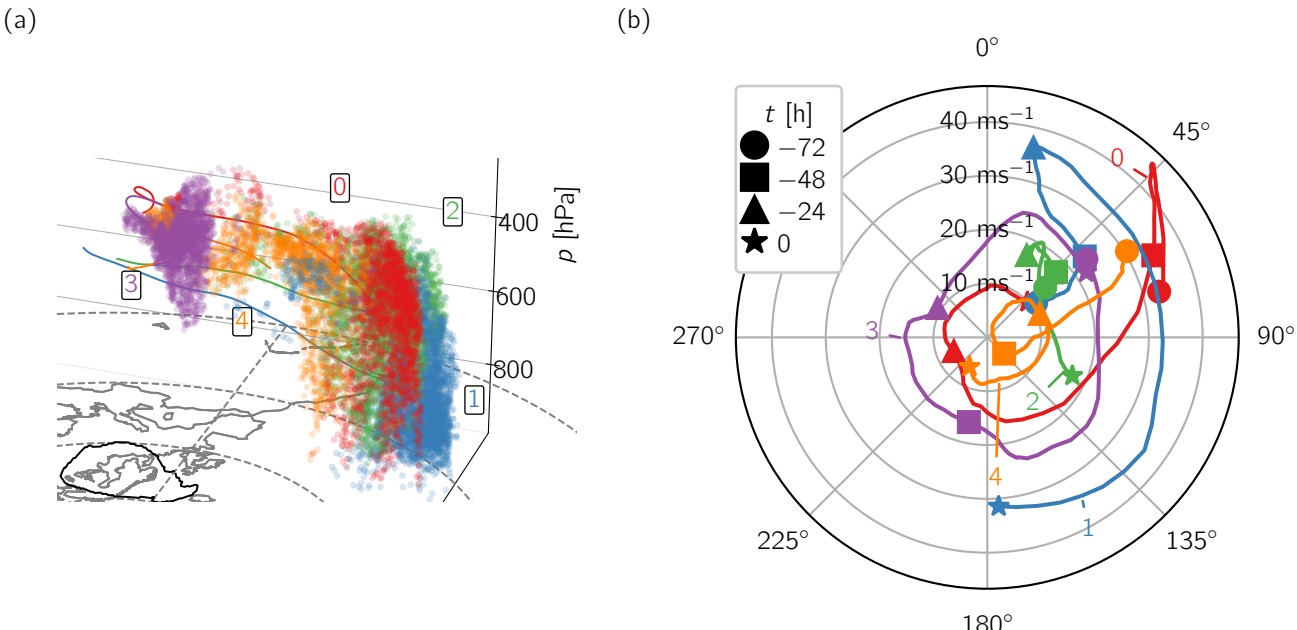

**Figure 11.** (a): Same as Fig. 5 but for backward trajectories initialized at 26 January 2017 18 UTC. Clustered trajectories for $\epsilon = 2 \times 10^4$ km$^2$. Points are shown for $t = -72$ h, black contour shows location of identified blocking regions at $t = 0$ h. One boundary cluster has been removed. (b): Median horizontal velocities of each cluster displayed as a hodograph. $0°$ corresponds to exclusively northward motion. Markers indicate time steps.

where the horizontal velocity components are displayed in a hodograph for all clusters shown in Fig. 11 (a). The polar plot enables display of both horizontal velocity components in one graph and makes clear the circular movement of the purple cluster. The orange (4) and red (0) clusters' arrival in the blocking can be similarly identified by their anticyclonic movement

later on, though the curves are a lot less clear. A decrease of zonal velocities reveals that the orange cluster arrives second before the red, the green (2) and finally the blue.

Tracking of the individual clusters confirms the hypothesis formulated in Sect. 4.2.1. The size of the blocking determines when and how long the traced parcels are inside its region of influence and, thus, how dense and how three-dimensional the parcels assemble at a particular point in time. The present example also demonstrates that the processes discussed do not occur

exclusively, but can coexist and contribute to the complexity of both individual cases and case-to-case variability.

## 5 Conclusions

The present study aimed to investigate the occurrence of warm conveyor belts (WCBs) feeding into atmospheric blocking adhering to the notion of spatial coherence. This promised to provide a complementing perspective on WCBs as phenomena



of first order importance for the behavior and development of atmospheric blocking. We also use this case study to prove the methodological value of the mathematical concept of coherent sets, which is an active area of research. To the best of our knowledge, neither the study of WCBs based on their spatial coherence nor the identification of coherent air streams in high-resolution three-dimensional atmospheric trajectories has been carried out thus far.

The adaptation, implementation and advancement of the methodology of Banisch and Koltai (2017) is the core contribution of this work. The developed framework respects the scale dissimilarities of atmospheric motion, handles boundary conditions in a physically consistent manner and provides flexibility for application to geoscientific phenomena on other scales all while keeping numerical efficiency and ease of application in mind. Broadening the range of phenomena studied in this manner is a subject of future work. To facilitate this, we provide a user-friendly `python` package, `GeoCS` (cf. code and data availability).

The case studies presented demonstrate our method's capability in finding WCBs based solely on spatial information. Apart from respecting the (arguably) common notion of WCBs as coherent, synoptic scale air streams, this perspective has the advantage of being able to conceive WCBs as spatio-temporally extended objects in time similar to atmospheric blockings. More specifically, we found that, on the one hand, air parcels that reach a blocking at a common point in time may have been part of the same or different WCBs and that, on the other hand, individual WCBs can contain air parcels that reach the same blocking at different points in time (cf. Fig. 6).

The WCBs found in our case studies were generally remarkably distinct from other coherent sets identified, both with respect to their dynamical properties and their pathways. Nevertheless, we did not find a rigorous dichotomy between moist and dry air streams as it might seem when looking at distributions of maximal potential temperature change over all trajectories traced over the whole lifetime of a blocking event (Steinfeld et al., 2022; Steinfeld, 2019; Pfahl et al., 2015). In agreement with existing research, for both of the two blockings investigated, the influence of WCBs was episodic and larger in earlier stages of the life cycle. Our results suggest that this was due to large scale subsidence, inavailability of moisture and self-containment of air parcels in blockings later in their life cycle. In earlier stages, however, we found that up to 100 % of the air parcels traced were influenced by a WCB at some point.

In addition to clustering trajectories, the presented method provides concise information on the shape of the point cloud represented by the air parcels traced at each point in time. In particular, we introduced the heuristics $d$ and $\ell$ approximating the dimension and grid length of the point cloud and emphasize that both of these behave considerably different than linear measures of distances between the points. Given that Lagrangian analysis is ubiquitous especially in atmospheric sciences, we think that these tools are a valuable contribution.

Applying these heuristics to our case studies showed that the stabilizing effect a blocking has on the synoptic flow field imprints on the coherence of trajectories that flow through it. This happens due to quasi-stationarity, reduced shearing and generally lower velocities. Consistently, strong and abrupt changes in the flow field, such as when a zonal regime reestablishes after a blocking, lead to shearing and filamentation of point clouds.

Given the complexity of the algorithm developed, a climatological evaluation of the effects discussed for the present case studies is non-trivial and, thus, subject of future studies. As mentioned above, we are convinced that the methodological framework presented will give valuable insights for other meteorological (or oceanic) phenomena such as cyclones.



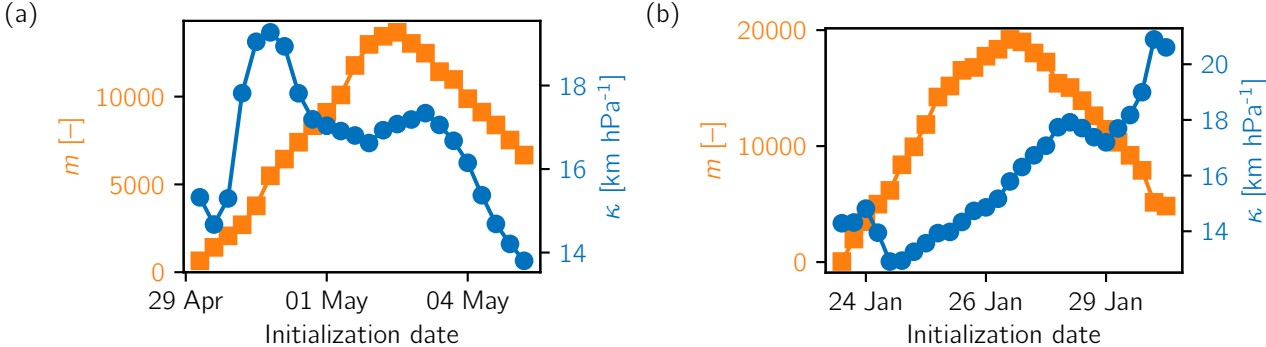

**Figure A1.** Number of trajectories $m$ (orange squares; left axes) and scale factor $\kappa$ (blue circles; right axes) calculated over all sets of trajectories and time steps for three-day forward and three-day backward trajectories at each initialization date.

*Code and data availability.* Code to reproduce the findings in this paper is published as a `python` package on the python package index
repository (pypi.org/project/GeoCS/). It also contains a list of all dependencies used. ERA5 reanalysis data is available from the ECMWF's climate data store (CDS; cds.climate.copernicus.eu). Trajectories and averaged diffusion operator matrices ($\mathbf{Q}_\epsilon$) can be provided upon request.

*Video supplement.* Videos of synoptic conditions for both case studies are provided in the supplement material along with several supplementary figures.

# Appendix A: Additional figures

*Author contributions.* All authors contributed to the conceptual framework of the paper. HS carried out the computations and was responsible for creating the figures and drafting the paper. RC contributed to Sect. 2. All authors contributed to revision of the manuscript.

*Competing interests.* The authors declare no competing interest.

*Acknowledgements.* This research has been funded by Deutsche Forschungsgemeinschaft (DFG) through grant CRC 1114 "Scaling Cascades
in Complex Systems", Project Number 235221301, Project A08 "Characterization and Prediction of Quasi-Stationary Atmospheric States". Furthermore, the authors like to thank Daniel Steinfeld for providing help in analysing trajectories. Numerous colleagues provided helpful comments throughout the production of the paper.





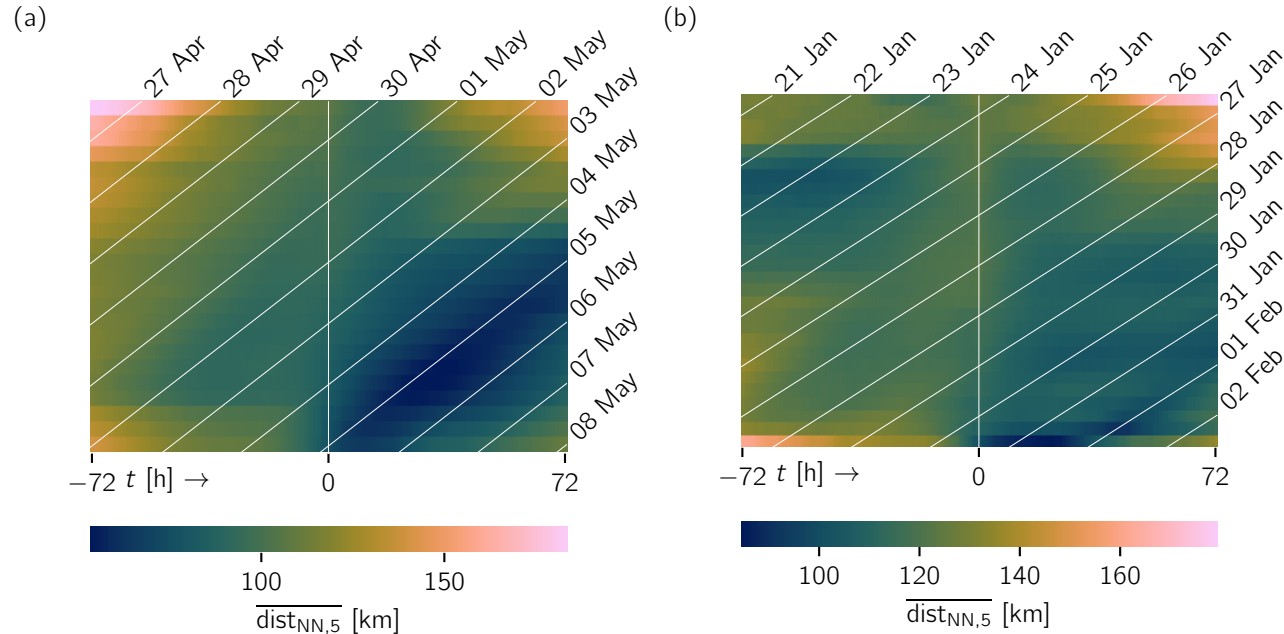

**Figure A2.** Same as Fig. 4 (a), but for the mean of the five nearest neighbor distances for each point for (a) the 2016 case and (b) the 2017 case.

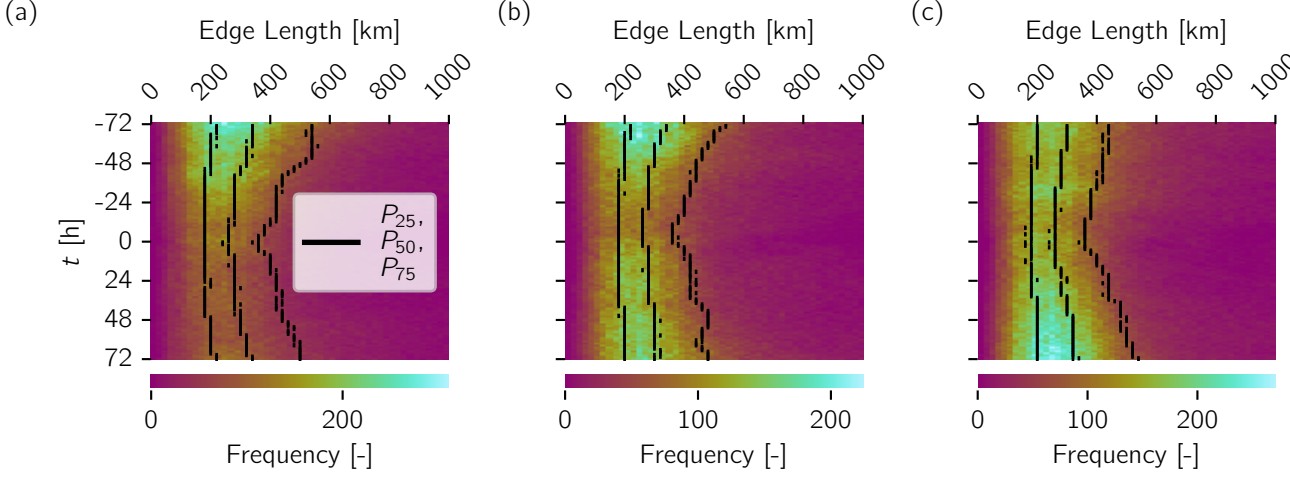

**Figure A3.** 2D histogram of edge lengths of the $\alpha$-hulls generated by $\alpha$-shapes for the trajectories initialized at (a) 02 May 2016 00 UTC, (b) 04 May 2016 00 UTC and (c) 26 January 2017 18 UTC. Black lines indicate median and quartiles of the distributions for a specific $t$.



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
