# Peer review of "Assessing Lagrangian Coherence in Atmospheric Blocking"

_EGUsphere, 2024_

## Author Response (AR1)

**1 Referee 1**

This paper describes the use of a technique, developed by extending and adapting the procedures in Banish and Koltai (2017) to the analysis of atmospheric blocking. Two particular study cases are analysed in detail. The paper is interesting, well written, and represents a valuable contribution to the characterization and understanding of blocking events. Particularly interesting is the Lagrangian identification and three-dimensional characterization of warm conveyor belts, and its influence on blocking formation.

The methodology, as recognized in the paper, is certainly complex. Fortunately, the authors provide Python code implementing it, available via `pypi.org/project/GeoCS/`. In general, the procedures are described in detail and established with rigour. There are many tunable parameters for which choices have to be made (alpha, kappa, r, epsilon, . . . ), which are justified by the authors. In my opinion, however, there is still one point in the methodology that remains weakly justified: One crucial point in the identification of coherent sets is the choice of the eigenvectors selected to perform the clustering. In all cases (e.g. Fig. 5a) the spectral gap after which eigenvalues are neglected is very weak, as the authors recognize. Thus, still there is some doubt about the extent on which the identified air masses can be called 'coherent', and what would be the difference if other choice of eigenvalues is done. I propose the authors to give a quantitative measure of the 'coherence' attained by all or of some of the detected sets (at least the most relevant, and for some choice of initial and final time) by assessing to which extent Eq. (1) is really satisfied by the sets (at least to some level of approximation). I imagine several more or less direct ways to check Eq. (1), although other quantitative assessments of coherence can also be given in terms of Cheeger ratios (Froyland, 2015) or other metrics (Froyland, 2013) that are expected to be optimized. In summary, I think the paper would be suitable for publication if the authors provide some extra evidence that the detected sets are really 'coherent' or at least significantly more 'coherent' than sets evolved, for example, from initial patches selected just by spatial proximity.

**Response:** Thank you for your helpful advise. We agree that a measure of coherence of the sets identified by our method was lacking in the initial version of our paper. We have therefore developed such a metric that was derived in a natural way from our methodological setting. More specifically, we calculate time-averaged exit probabilties from individual sets based on transition probabilities in Q_epsilon. We compare the coherent sets found to test sets sampled randomly by spatial proximity and find the clusters identified are significantly more coherent.

The resulting changes in the manuscript are the inclusion of subsection 2.4 in the theory section of the manuscript as well as figures discussing the results when applied to the case studies in the respective sections 4.1.2 and 4.2.2 including the appropriate figures. A reference to this addition has also been made in the conclusion section and a figure was added to the supplement.

Other minor points:

- The bibliography is rather complete. However I think there are still a few papers related to this topic that consider either coherence or diabatic heating in blockings from the Lagrangian point of view and that can complete the reference list and being properly cited. Among them I suggest: Ehstand, N. et al.: Characteristic signatures of Northern Hemisphere blocking events in a Lagrangian flow network representation of the atmospheric circulation, Chaos 31, 093128 (2021). https://doi.org/10.1063/5.0057409. Zschenderlein, P. et al.: A Lagrangian analysis of upper-tropospheric anticyclones associated with heat waves in Europe, Weather Clim. Dynam., 1, 191–206 (2020). https://doi.org/10.5194/wcd-1-191-2020.

**Response:** We have made references to both of the suggested publications, which we agreed were relevant for our work. In particular, the former publication supported our observation that atmospheric blockings suppress synoptic scale dispersion of air parcels, which is a result of our discussion of the derived quantities $d$ and $\ell$. The second publication studies the interaction between European heat waves and atmospheric blocking with trajectory-based methods. The authors raise awareness to the fact that cross-isentropic ascent into blockings may also be linked to convective processes, which we have, therefore, also commented on in the manuscript.

- page 7, line 191: please rewrite this sentence to clarify to which set the expression 'this boundary set' refers to.

**Response:** We have made more clear the difference between the sets identified by the alpha shape algorithm (we call those "boundary sets") and the sets identified by k-means clustering containing mostly boundary points (we call those "residual sets"). Adjustments have accordingly been made throughout the manuscript.

- page 13, line 367: I think 'placed in a regular three-dimensional grid' should be rather 'placed close to a regular three-dimensional grid', since Sect. 3.2 states that some random displacements are applied.

**Response:** We have modified the statement to "approximately in a regular grid..."

- page 17, line 441: $\mathbf{P}_{\epsilon,t}$ is substochastic, not stochastic.

**Response:** This was a typo, we have since corrected.

- To help the readers, please indicate in the captions of Figures S3 and S4 the pertinence to the Canadian or to the European case of the different panels.

**Response:** We have done so.

**2 Referee 2**

The paper extends the diffusion map approach for the detection and characterization of coherent sets from tracer trajectories by Banisch & Koltai (2017) to the context of studying atmospheric blocking events. Two major blocking events are analyzed and the extracted coherent sets are shown to relate to warm conveyor belts, which are known to play a crucial part in the stabilization of the blocking regime. The proposed Lagrangian framework for the study of blocking events is novel and certainly of interest to the readers of NPG. The paper is very well written. Helpful graphics illustrate the careful computational studies, with all parameter choices well justified by the authors. However, I fully agree with the points raised by the other referee: In particular, a quantitative confirmation of the degree of coherence of the extracted flow patterns would considerably strengthen the study and should definitely be pursued.

**Response:** The issue of coherence quantification has been commented on above.

Minor points:
– Figure 1: It is unclear at this point what "2 pvu" means (pvu is only introduced later). Abbreviations should be explained in the caption. What is the meaning of the black arrow?

**Response:** We have elaborated the caption.

– Introduction: For readers unfamiliar with the underlying mechanisms leading to atmospheric blocking, some brief explanation on the meaning of negative potential vorticity anomaly would be helpful.

**Response:** We have added a sentence about negative potential vorticity anomalies.

– 281: "vertical distance of 7 hPa between 550 and 150 hPa". Can one say anything about the approximate altitude of these levels?

**Response:** We have added that this encompasses the upper troposphere (whose height above the ground varies significantly).

– 291: capitalized R: "LAGRangian ANalysis TOol (LAGRANTO)"
**Response:** Done.

– 301: Definitions of $u_t^i, v_t^i, w_t^i$ would be helpful.

**Response:** We have adjusted the definition.

– Figure 2: The tropopause is difficult to distinguish from the continental borders. Please plot this in a different color. Insertion of the dateline could be helpful, because it is mentioned in the main text (334).

**Response:** We have changed the plot to indicate the tropopause by a dotted line. The dateline is already included.

– 377: "indicate" instead of "indicated"

**Response:** Done.

– Figure 5a): The position of the epsilon-legend is suboptimal as the dots are not really separated from the plotted eigenvalues.

**Response:** We have adapted the order of the epsilon values to prevent this issue from happening. While we were at it, we also added values for epsilon=1e6, since one can argue that the curves in Figure 3 still appear linear at those values.

– Figure 5b) Do the gray points correspond to the boundary set? This should be mentioned in the caption.

**Response:** As mentioned above, we have adapted our terminology wrt "boundary" and "residual" sets. In addition, the respective caption has been modified to include reference to the residual set.

– 441: "substochastic" instead of "stochastic". Is the spectral gap after the sixth instead or after the seventh eigenvalue (later the first six eigenvectors are used for clustering)?

**Response:** There was indeed a bit of confusion wrt the location of the spectral gap and the number of cluster to be chosen for k-means clustering. The revised manuscript is now consistent in this regard. For the first application (02 May 2016 00UTC), there was only a typo, the clusters didnt change. For the second application (04 May 2016 00UTC), we indeed used an incorrect number of clusters. The additional cluster identified by k-means is clearly a residual cluster. The interpretation of the physical processes determining the behaviour of the air streams didnt change (cf. the according figures). As also mentioned in the manuscript, the results are quite robust to variation of $k$ and $\epsilon$. In the last application (26 January 2017 18UTC), we again used one cluster too few. The additional cluster (purple in the revised manuscript) is made up of parcels that used to belong to adjacent clusters. Again the interpretations haven't changed.

– Figures 5 + 7: Cluster coloring is different.

**Response:** We have made the choice of colour more consistent across case studies. Specifically, the residual cluster is now always gray. As mentioned also in the manuscript, the labels of the clusters are arbitrary and so are the colours. There is no hierarchy between the clusters from the perspective of the k-means algorithm.

– 523: "the brown": there is no brown cluster with number 0 in figure 7.
**Response:** The text now agrees with the figures.

– Figures 7– : I find captions starting with "Same as Fig ..." very inconvenient. Please insert the details, no matter if this causes repetitions.
**Response:** We have revised the captions.

**3   Additional Changes**

- added mention of warm conveyor belts in the abstract

- figures of $d$ indicate that $d$ is unitless

- several figure compositions have been reworked to accout for the additional exit probability figures

- In the WCB plots over lifecycle the WCB fraction has now consistent unitless indication ([–] instead of [-]) and number of trajectories is labelled consistently with $m$ instead of $m_i$

- Hodograph has been reworked for better readability

- The $\alpha$-hull edge length histograms in the Appendix (Figure A3) labels quartiles with $Q_i$ now to avoid confusion with $P_{\text{exit}}$

- The supplement video has been uploaded to TIB AV-Portal as per recommendation by Copernicus.

- Acknowledgement of authors of python packages used has been added